# Comprehensive Tools of Alkaloid/Volatile Compounds–Metabolomics and DNA Profiles: Bioassay-Role-Guided Differentiation Process of Six *Annona* sp. Grown in Egypt as Anticancer Therapy

**DOI:** 10.3390/ph17010103

**Published:** 2024-01-11

**Authors:** Mona A. Mohammed, Nahla Elzefzafy, Manal F. El-Khadragy, Abdulhakeem Alzahrani, Hany Mohamed Yehia, Piotr Kachlicki

**Affiliations:** 1Medicinal and Aromatic Plants Research Department, Pharmaceutical and Drugs Industries Institute, National Research Centre, Dokki, Giza 12622, Egypt; 2Cancer Biology Department, National Cancer Institute, Cairo University, Cairo 11976, Egypt; nahla.elzefzafy@gmail.com; 3Biology Department, Faculty of Science, Princess Nourah bint Abdulrahman University, P.O. Box 84428, Riyadh 11671, Saudi Arabia; mfelkhadragy@pnu.edu.sa; 4Food Science and Nutrition Department, College of Food and Agricultural Sciences, King Saud University, P.O. Box 2460, Riyadh 11451, Saudi Arabia; aabdulhakeem@ksu.edu.sa (A.A.); hanyehia@ksu.edu.sa (H.M.Y.); 5Department of Food Science and Nutrition, Faculty of Home Economics, Helwan University, Helwan 11611, Egypt; 6Institute of Plant Genetics, Polish Academy of Sciences, 60-479 Poznan, Poland; pkac@igr.poznan.pl

**Keywords:** metabolomics, fingerprint profiles, bioassay-guided differentiation, volatile oils, *Annona* sp., antiproliferative agent

## Abstract

Trees of the *Annona* species that grow in the tropics and subtropics contain compounds that are highly valuable for pharmacological research and medication development and have anticancer, antioxidant, and migratory properties. Metabolomics was used to functionally characterize natural products and to distinguish differences between varieties. Natural products are therefore bioactive-marked and highly respected in the field of drug innovation. Our study aimed to evaluate the interrelationships among six *Annona* species. By utilizing six Start Codon Targeted (SCoT) and six Inter Simple Sequence Repeat (ISSR) primers for DNA fingerprinting, we discovered polymorphism percentages of 45.16 and 35.29%, respectively. The comparison of the profiles of 78 distinct volatile oil compounds in six *Annona* species was accomplished through the utilization of GC-MS-based plant metabolomics. Additionally, the differentiation process of 74 characterized alkaloid compound metabolomics was conducted through a structural analysis using HPLC-ESI-MS^n^ and UPLC-HESI-MS/MS, and antiproliferative activities were assessed on five in vitro cell lines. High-throughput, low-sensitivity LC/MS-based metabolomics has facilitated comprehensive examinations of alterations in secondary metabolites through the utilization of bioassay-guided differentiation processes. This has been accomplished by employing twenty-four extracts derived from six distinct *Annona* species, which were subjected to in vitro evaluation. The primary objective of this evaluation was to investigate the IC_50_ profile as well as the antioxidant and migration activities. It should be noted, however, that these investigations were exclusively conducted utilizing the most potent extracts. These extracts were thoroughly examined on both the HepG2 and Caco cell lines to elucidate their potential anticancer effects. In vitro tests on cell cultures showed a significant concentration cytotoxic effect on all cell lines (HepG2, HCT, Caco, Mcf-7, and T47D) treated with six essential oil samples at the exposure time (48 h). Therefore, they showed remarkable antioxidant activity with simultaneous cytotoxic effects. In total, 50% and 80% of the A. muricata extract, the extract with the highest migratory activity, demonstrated a dose-dependent inhibition of migration. It was strong on highly metastatic Caco cells 48 h after treatment and scraping the Caco cell sheet, with the best reduction in the migration of HepG2 cells caused by the 50% *A. reticulata* extract. Also, the samples showing a significant IC_50_ value showed a significant effect in stopping metastasis and invasion of various cancer cell lines, making them an interesting topic for further research.

## 1. Introduction

The global prevalence of cancer is on the rise. By the year 2023, it was projected that around 1,918,030 individuals in the United States will receive a new cancer diagnosis, and 609,360 individuals will succumb to the disease. The mortality rate for cancer is greater in men compared to women, with 196.8 deaths per 100,000 men and 139.6 deaths per 100,000 women [1]. The most prevalent types of cancer include endometrial cancer, pancreatic cancer, thyroid cancer, liver cancer, lung and bronchus cancer, prostate cancer, colon and rectum cancer, skin melanoma, bladder cancer, non-Hodgkin lymphoma, kidney and renal pelvis cancer, and melanoma of the bladder [1]. By 2030, the annual incidence of cancer is projected to reach 23.6 million cases [1]. Over 60% of the global population opts for traditional medicine as their primary approach in addressing various health ailments, including cancer. Traditional herbal remedies have been utilized for centuries and continue to be employed today [2,3]. Nevertheless, cancer treatment is intricate, and the present outlook for patients is contingent upon factors such as their age, gender, and overall well-being, as well as the specific type and stage of their illness.

The efficacy of chemotherapy in the initial phases of cancer is often high, although it is contingent upon the patient’s physiological condition and the prescribed medicine regimen. Moreover, the identification of anticancer medications has tremendously profited from the copiousness of chemical compositions present in natural substances. Approximately 49% of the chemotherapy drugs utilized in the domain of oncology pharmaceutics are either derived from or influenced by natural sources [4]. Examples of these chemicals include anthracyclines, podophyllotoxins, and etoposides, which are topoisomerase inhibitors. Additionally, taxanes, vinca alkaloids, and other tubulin-binding drugs are also included [5,6]. These examples demonstrate the potential of natural ingredients in the field of pharmaceutical research.

The role of nature as a significant contributor to the progress of anticancer therapies should not be underestimated. Numerous cytotoxic medications currently employed in clinical settings are derived from plants and other natural origins. The induction of various forms of human cancer is directly linked to the presence of reactive oxygen species (ROS), necessitating the use of antioxidants or scavengers to neutralize their effects [7].

The genus *Annona* is one of 129 genera of the Annonaceae family and contains 119 species (trees and shrubs) with eight species grown in Egypt for commercial uses. Egypt produces from 1100 to 1082 tons of *Annona* [8]. The most important species for fruit production are Sweetsop or sugar apple (cvs. *Annona squamosa*; Balady and *abdel-razek*), Soursop (*Annona muricata*), and Cherimoya (*Annona cherimola*; Hindi cv.) [9,10]. The plant is conventionally employed for the management of dysentery, cardiac ailments, epilepsy, diarrhea, fever, pain, rheumatism, and arthritis, worm infestation, constipation, hemorrhage, antibacterial infection, and antiulcer purposes. Additionally, it possesses antifertility and antitumor characteristics; its leaves are utilized for diabetes, headaches, antidepressants, antileishmanial purposes, and insomnia [11,12,13]. 

The *Annona* genus is part of the Annonaceae family, which consists of 129 genera. It comprises 119 species, including both trees and shrubs. In Egypt, eight of these species are cultivated for commercial purposes. Egypt’s annual *Annona* production ranges from 1100 to 1082 tons [8]. The key species for fruit production include Sweetsop or sugar apple (cvs. *Annona squamosa*; Balady and *abdel-razek*), Soursop (*Annona muricata*), and Cherimoya (*Annona cherimola*; Hindi cv.) [9,10]. The herb is traditionally used to treat dysentery, heart conditions, epilepsy, diarrhea, fever, pain, rheumatism, arthritis, worm infestations, constipation, hemorrhage, bacterial infections, and ulcers. In addition, it has antifertility and anticancer properties. Its leaves are used for treating diabetes, headaches, depression, leishmaniasis, and sleeplessness [11,12,13]. The therapeutic significance of the *Annona* species trees is attributed to the existence of certain distinct secondary metabolites such as alkaloids (asisoquinoline, aporphine, proaporphine, and oxoaporphine groups) [14], glycosides, terpenes, cyclopeptides, flavonoids, resins, volatile oils, tannins, and acetognins, among others [15]. Moreover, *A. reticulata* has been observed to be potentially employed as a chemopreventive agent in cancer therapy. Furthermore, *A. muricata* is a well-known medicinal remedy in Africa, America, and India for cancer treatment [16]. *A. squamosa* and *A. atemoya* are employed in the treatment of tumors [17]. *A. abdel-razek* is an Egyptian cultivar resulting from a cross between *A. cherimola* and *A. squamosa*. Recently, it has also been cultivated in other Arabian countries including Sudan, Lebanon, Oman, Kuwait, Palestine, and Jordan [18].

Metabolic profiling approaches can be easily applied to compare secondary metabolites utilizing modern bioinformatics software tools, which may eventually determine all metabolites for *Annona* sp., simplifying mounting and targeting of bioactive compounds. Metabolomics is a thorough study of metabolic reactions that includes DNA fingerprinting (footprinting) and targeting or profiling secondary metabolites to detect and correlate metabolites in a cell or organism. Fingerprinting tries to obtain a “chemical picture” of the sample, where the signals cannot always be used to detect or identify specific novel metabolites and are heavily dependent on the sophisticated technique used [19].

Due to the expansion of *Annona abdel-razek* cultivation and its increasing consumption in the main Egyptian and export markets, it is necessary to investigate the chemical composition of this plant in comparison to other cultivar species. In the current paper, we report the results of the first metabolomics research for *Annona* species through tentative identification of 75 alkaloids using LC/MSMS methods and fingerprinting, then using a bioassay-guided differentiation process as an antiproliferative agent. This study aims to search for physiologically active molecules in fractions from six *Annona* species. These cytotoxicity investigations will provide a pathway to the identification of new, non-toxic natural cures, and the lead compound will be a candidate for the synthesis of more biologically active chemicals by chemical means or enzymatic transformations from the Egyptian species *Annona* to test its antitumor potential using a cancer cell line model.

## 2. Results

### 2.1. Extraction of Plant Material and Isolation of DNA, Followed by Fingerprinting Analysis

The current study included aerial parts of six different genotypes of *Annona* sp., including *A. atemoya*; *A. glabra*; *A. muricata*; *A. reticulate*; *A. squamosa*; and *A. abdel-razek*, that were gathered in the Giza governorate of Egypt (in Figure 1). The total genomic DNA was isolated from young leaves of *Annona* sp. greenhouse-grown plants according to the CTAB protocol [20]. SCoT and ISSR amplification was performed as described in [21], using 12 primers (Table 1).

### 2.2. Bioassay-Guided Differentiation Process for Six Annona Species and Isolated Total Alkaloids

Figure 2 depicts the sequential extraction techniques (50%, 80%, and 100% methanol) used to extract six different species of *Annona* plants. The extraction technique involved utilizing 100 mL of a solvent to extract 25 g of dry plant material. This process was performed three times, with each extraction accompanied by agitation at a speed of 170 revolutions per minute. The results of these extractions showed differences in the weights obtained for each species.

### 2.3. Taxonomic—DNA Fingerprinting for Six Annona Species

The SCoT and ISSR banding profiles produced by the twelve 10-mer primers in the six samples of *Annona* species are illustrated in Figure 3 and Figure 4 for primers.

From Figure 3, SCoT (1–4, 6, and 8) showed 45.16% polymorphisms and 17 monomorphisms from six SCoT, giving 31 total bands; these data were then analyzed using the SPSS program shown in Figure 5A. The dendrogram of SCoT found complete similarity between *A. glabra* and *muricata*, then *atemoya*, then *reticulata*, then *squamosa*, then *abdel-razek* (Appendix A). The Figure 4 HB-12 primer showed eight bands, giving five monomorphic and three polymorphic bands, resulting in 37.5% polymorphisms, while HB-10 showed six bands, giving two monomorphic bands and four polymorphic bands, resulting in 66.66% polymorphisms. The dendrogram of the ISSR analysis for six species showed a series of similarity as follows: *atemoya* > *abdel-razek* > *reticulata* > *glabra* > *squamosa* > *muricata* (Figure 5B). A combination of SCoT data and ISSR data gives in Appendix A and Figure 5C the similarity between the SCoT and ISSR analysis for six species as follows: *atemoya* > *reticulata* > *glabra* > *muricata* > *squamosa* > *abdel-razek*. 

### 2.4. Volatile Oils of Annona sp. and Chemical Characterization Using GC-MS

As shown in Table 2, the aerial part of the six species of *Annona* sp. contained the chemical constituents, and the highest amount of essential oil was in *A. abdel-razek* (0.305), then *muricata* (0.15), *atemoya* (0.071), *squamosa* (0.055), and *reticulata* (0.028) of mL/25 g fresh. A statistical analysis showed significant differences in oil content between the six species of *Annona* sp. From the GC-MS analysis, the identified constituents of the volatile oil are presented in Table 2 and Figure 6. Based on their retention indices and mass fragment patterns concerning the NIST 08 and Wiley version 1.2 databases, the aerial sections of *Annona* sp. that contained volatile aromatic hydrocarbons were identified. As oxygenated monoterpenes, sesquiterpene hydrocarbons, oxygenated sesquiterpenes, and oxygenated diterpenes, a total of 76 aromatic volatile elements account for 100% of the total oil contents of all species. The main constituents of the volatile oil identified with GC-MS of six species are β-pinene and α-pinene (8.43% and 8.03% in *abdel-razek*, and 9.45% and 9.1% in *muricata*, respectively), caryphyllene (28.09, 19.59, 5.37, 11.66, 16.63, and 6.22 in *atemoya*, *glabra*, *abdel-razek*, *reticulata*, *squamosa*, and *muricata*, respectively), and germancrene D (9.5, 4.98, 13.06, 10.74, 4.47, and 6.32 in *atemoya*, *glabra*, *Abdel-razek*, *reticulata*, *squamosa*, and *muricata*, respectively).

### 2.5. LC/MSMS Profiles of Six Annona sp. with 18 Differentiation Extracts

Metabolomic analyses of bioassay-guided differentiation process methanolic *Annona* sp. extracts were performed using high-resolution mass spectrometry (HR-MS) systems on targeted MS/MS acquisition. The metabolites of Type’s alkaloids were examined using HR-MS in a data-dependent acquisition (DDA) analytical positive mode. For all metabolites, the automated acquisition of MS/MS spectra was found (MS2). The analysis of molecular networking (MN) was performed in conjunction with the prioritization of bioactivity and the profiling of metabolites in a large series of extracts from the *Annona* sp. plant. The basis of such an acquisition mode is the injection of a given sample three times in repetition, and the connection between data processing and data acquisition can enable the adjustment of MS/MS acquisition parameters to attain virtually full MS/MS sampling coverage in approximately real-time. It was shown that the DDA approach generates significantly more MS/MS events than traditional DDA by temporally separating data processing from acquisition, thereby maximizing the data acquisition time during the chromatographic gradient by minimizing the competing processing time, using such an approach on a complex mixture with HPLC-ESI-MS^n^ and UPLC-HESI-MS/MS. It will be interesting to see how this area develops in the future [22]. The acquired LC/MS data were independently processed for positive ionization using MS-DIAL ver. 4.60. Following each scan of the raw data files, lists of masses were produced in the first stage. After that, the chromatogram Builder algorithm was created for each mass regularly observed throughout the scans. After gap filling, isotope removal, compound and complex searches, and retention time normalization among peak lists, peak Table 3 was created. The generated tables were then submitted to MetaboAnalyst for the statistical analysis after the resultant data were log converted and filtered through interquartile range calculation to remove variables that have nearly constant values across the experiment settings housekeeping [23]. From Figure 6, the purpose of building the PLS-DA model and dendrogram involves the similarity between species as *abdel-razek* > *squamosa*, *reticulata* > *muricata*, and *atemoya* > *glabra* in Figure 7. Figure 8 shows a PLS-DA permutation result. Figure 8 shows the PLS-DA loading plot of different extract species. Hierarchical clustering of all signals from different *Annona* species is shown in positive clusters.

The LC-MSMS profile was used as a marker for the antiproliferative agents. *Annona* plants can produce diverse bioactive alkaloids. The precise molecular masses (less than 5 ppm), mass spectra, and retention times of the individual compounds were compared to those of the reference compounds available in PubChem, ChEBI, Metlin, KNApSAck, and literature data. When several compounds are co-eluted from the LC columns, the mass spectrometer is unable to analyze and fragment each component separately. Two different strategies have been used to overcome these data: (i) extending the HPLC-MS^n^ experiments’ separation period to 1 h, and (ii) using the C18 reversed-phase UPLC-HESI-MSMS to analyze the fractions acquired from the polyamide preparative LC run. Due to the design of the experiment, it was possible to identify chemicals that were only minimally present in plant tissues and were being obscured by their quantitatively dominant metabolites. Different alkaloid-type classes of 74 secondary metabolites involved tentative identification from the bioassay-guided differentiation process and methanolic *Annona* sp. Aporphine types as major groups in *Annona* sp. included Ocoteine, Apomorphine, Asimilobine, Thaliporphine, Nuciferoline, Lirinine-N-oxide, N-acetyl-3-methoxynornantenine, Domesticine, Magnoflorine, Bracteoline, Romucosine A, Isoboldine, Launobine, Predicentrine, Liridinine, Glaucine O,O-dimethylisoboldine, N-methylcorydine B, and others shown in Table 3. The oxoaporphine alkaloid type gave Annonbraine, Oxoanolobine, Dehydrocrebanine, Artabotrine, Liriodenine, Lanuginosine, Lysicamine, Oxolaureline, N-acetylbongaridine. The proaporphine alkaloid type involved (−)-N-formylstepharine, N-acetylstepharine, and Stepharine. Benzylisoquinoline- and isoquinoline-type alkaloids are shown as 7-O-methylcoclaurine, Reticuline, Coclaurine, N-methylcoclaurine, N,N-Dimethylcoclaurine, Annocherine A, Anomoline, and Annosqualine as shown in Table 4 and Figure 9.

#### 2.5.1. Mass Spectral Analysis of Norisoboldinedemethyl

The MS^n^ fragmentation pattern of norisoboldine demethyl was examined to aid in a better understanding of the MS^n^ spectra of its metabolites. The protonated molecular ion showed a predominant ion at *m*/*z* 314.1385 (Figure 10A). The MS2 spectra of the ion at *m*/*z* 314.1385 displayed two major product ions at *m*/*z* 265.0862 (Figure 10A). The MS2 product ion at *m*/*z* 298.8 led to an MS^3^ product ion at *m*/*z* 264.6 (Figure 10B). The ion at *m*/*z* 298.8 was probably formed via the loss of the amino group, as indicated in pathway a (Figure 10B). The other ion at *m*/*z* 265 was formed via the loss of a molecule of methanol from [M + H − NH^3^]^+^. The detection of these characteristic product ions from low and high resolution suggests these are the C_18_H_19_NO_4_ chemical formula metabolites [69]. Interestingly, the main extracts 50% *abdel-razek* and 50% *A. reticulata* were the most high and more than 13 other extracts’ accumulation of norisoboldinedemethyl shown in Figure 10C.

#### 2.5.2. Mass Spectral Analysis of N,N-Dimethylcoclaurie

Benzyl-isoquinolines presented the loss of NH_3_ (coclaurine), CH_3_NH_2_ (*N*-methylcoclaurine), or C_2_H_6_NH (*N,N*-dimethylcoclaurine) at *m/z* 265.0876, substituents of the isoquinoline core, and also shared a common ion at *m/z* 121.0651 (C_8_H_9_O^+^), which corresponds to the acetophenone moiety generated by inductive cleavage [25]. Finally, spirobenzylisoquinolines showed not only the neutral loss of substituents of the alkaloid core but also the breakage at the isoquinoline core, generating ions at *m/z* 175.0755 (C_11_H_11_O_2_) (2-benzylbut-3-enoic acid) (in Figure 11A,B). The highest concentrations in 50% extract *abdel-razek* and 80% extract *A. reticulata* were more than 13 other differentiation extracts’ accumulation of *N,N*-dimethylcoclaurie shown in Figure 11C.

#### 2.5.3. Mass Spectral Analysis of Magnoflorine

Magnoflorine is an aporphine alkaloid. It had mass spectra essentially identical to the standard, with a base peak of *m*/*z* 342.1696 (Figure 12A,B), corresponding to [M]^+^ (calc. 342.1700, C_20_H_24_NO_4_), accompanied by fragments of *m*/*z* 297.1120, resulting from the loss of NH(CH_3_)_2_, and of *m*/*z* 265.0863, C_17_H_13_O_3_, generated via further loss of MeOH, with these fragments. Interestingly, the main extract 50% *abdel-razek* was the most high and more than 16 other extracts’ accumulation of magnoflorine shown in Figure 12C. 

#### 2.5.4. Mass Spectral Analysis of Isoboldine

In the positive ion mode, isoboldine produced protonated molecular ions [M + H]^+^ at *m*/*z* 328.1542 and 279.1125 as the most intensive precursor ions, respectively. Therefore, [M + H]^+^ at *m*/*z* 328.0 was in low resolution (Figure 13B); isoboldine generated two major product ions at *m*/*z* 265.0860 and 297.1125 (Figure 13A). The transitions of *m*/*z* 297.1125–265.0860 for isoboldine had higher intensity than those of *m*/*z* 328.1541. So, the precursor/product ion pairs at *m*/*z* 328.1541/265 were selected for quantification of isoboldine in the MRM mode. The main extracts were 50% *abdel-razek* and 80% *A. reticulata,* with the highest accumulation of isoboldine among the other 13 extracts (Figure 13C).

### 2.6. Antiproliferative Agent for Six Annona Species Grown in Egypt

HepG2, T47D, MCF7, HCT, and Caco cells were incubated with a range of concentrations of six *Annona* species and their fractions to estimate the minimum half inhibitory concentration (IC_50_) of these compounds using an SRB assay. After 48 h of incubation, cell numbers and viability were subsequently reduced with dose escalation, as shown in Table 3 and Appendix A. Based on a GraphPad Prism analysis, the IC_50_ values were determined; we considered IC_50_ above 100 μg/mL as non-promising effects. Interestingly, the results showed that the *A. atemoya* and *A. squamosa* volatile oils have strong inhibitory effects against all the cell lines except T47D, while the *A. glabra* volatile oil has a very strong inhibitory effect against only both HepG2 and MCF7,18 μg/mL and 26.5 μg/mL, respectively. Interestingly, the observed volatile oil of *A. glabra* has a very strong effect against the MCF7 breast cancer cell line but has no effect against the T47D breast cancer cell lines. The same is true for the volatile oil of *A. abdel-razek*, which shows a slight inhibitory effect against only the MCF7 and Caco cell lines [12]. Moreover, the volatile oil of *A. muricata* shows a very specific and strong inhibitory effect against only colon cancer cell lines. Surprisingly, *A. reticulata* volatile oil shows no inhibitory effect against all the cell lines, while its varying eighteen concentrations of the alcoholic extract show a very strong and comparable inhibitory effect against different cell lines, leading to the conclusion that *A. reticulata* differentiation extract secondary metabolites have a synergistic effect between them. The extraction of various alcoholic extracts from different *Annona* species was found to be dependent on the synergistic effects of secondary metabolites. This extraction process resulted in a potent inhibitory effect, particularly in the case of three concentrations of *A. atemoya*, *A. glabra*, and *A. muricata* on the MCF7 cell line, with an observed inhibition of less than 20 μg/mL. Similarly, *A. glabra*, *A. abdel-razek*, and *A. reticulata* showed promising results on the HepG2 cell line, with an observed inhibition of less than 15 μg/mL. These findings strongly support the recommendation of these particular *Annona* species for further research on their cytotoxic effects. However, it is worth noting that these studies have not yet been approved for inclusion in clinical trials.

From the data presented in (Table 2), it can be inferred that the examined sample demonstrates a marked inhibitory effect on various malignant cells at varying concentrations of the alcoholic extract. These findings suggest that these extracts possess considerable potential in selectively inhibiting the growth of malignant cells, with each extract exhibiting a distinct mechanism of action. Additionally, subsets of the most potent four to eight extracts from each cell line were selected for a subsequent biochemical and molecular analysis. Malondialdehyde (MDA) serves as a radical oxidative marker, whereas Superoxide Dismutase (SOD) assumes the function of an endogenous antioxidant. It has been hypothesized that neoplastic cells experience heightened levels of oxidative stress in contrast to their normal cellular counterparts.

#### 2.6.1. Determination of Total Lipid Peroxide Content (Measured as Malonaldialdehyde)

Malondialdehyde (MDA) serves as the ultimate outcome resulting from the process of lipid oxidation. Thus, it is an indicator of cellular damage due to oxidative stress. HepG2 and T47D cells have been subjected to treatment with the most potent extracts derived from the *Annona* sp. on each line for a duration of 48 h. The alteration in the level of MDA, also known as lipid peroxidation, which serves as an indicator of oxidative stress, was assessed by measuring the content of MDA. In comparison to the untreated control cells, the MDA content in HepG2 cells treated with 50% *A. glabra* and 100% *A. glabra* extracts exhibited a significant increase, whereas the volatile oil *A. glabra* displayed a slight reduction in the MDA level. The other potent cytotoxic extracts resulted in a substantial decrease in the MDA level when compared to the untreated cells. In the case of T47D cells, treatment with different extracts for 48 h led to a significant modification in MDA activities when compared to the control, untreated cells. The most pronounced activity of malondialdehyde was observed when T47D cells were subjected to treatment with 50% *A. muricata* extracts in comparison to the control untreated cells. Conversely, 80% *A. atemoya* showed a slight increase in MDA activity compared to the control, while 80% *A. glabra* and 80% *A. reticulata* exhibited a decrease in MDA activity. Furthermore, 80% *A. muricata* displayed the lowest MDA activity in comparison to the control group and the other extracts. The HCT cell line was subjected to treatment with various extracts for a duration of 48 h, resulting in virtually no effect on MDA levels. However, 100% *A. muricata* exhibited a slight increase in MDA activity when compared to the other extracts and the control cells. The remaining extracts showed no effect on MDA levels in comparison to the control. Caco cells subjected to treatment with different extracts for 48 h demonstrated significant alterations in MDA levels. The volatile oils of *A. atemoya*, 100% *A. atemoya*, 80% *A. reticulata*, 80% *A. muricata*, and 100% *A. muricata* extracts resulted in a significant increase in MDA activity when compared to the untreated control group and the other extracts. Moreover, the extract of *A. atemoya* resulted in a significant decrease in the activity of MDA when compared to the control (Figure 14). The objective of this current study was to establish an accurate and precise technique for determining MDA through the use of a derivative of thiobarbituric acid that reacts with substances. In order to induce pharmacological oxidative stress in cell cultures, hydrogen peroxide (H_2_O_2_) and tert-butyl hydroperoxide (t-BOOH), secondary metabolites known to trigger oxidative stress, were utilized. The findings indicate that MDA, along with its corresponding thiobarbituric acid, can serve as a valid and reliable biomarker for lipid peroxidation in human HepG2, T47D, Caco, and HCT cancer cell lines.

#### 2.6.2. Determination of Reduced Glutathione Content

T47D cells, subjected to various extracts for a duration of 48 h, exhibited a substantial alteration in GSH activities as compared to the control group of cells that were not treated. The greatest level of GSH activity was observed in T47D cells treated with 80% *A. atemoya*, in comparison to the untreated control cells. On the other hand, the 50% *A. muricata* extract displayed a minor increase in GSH activity compared to the control. However, the 80% *A. glabra*, 80% *A. reticulata*, and 80% *A. muricata* extracts had no impact on GSH activity (Figure 15). After subjecting HepG2 cells to various extracts for a duration of 48 h, noticeable alterations in the levels of GSH were observed. The GSH activity showed a substantial reduction when treated with 100% *A. glabra* and 100% *A. reticulata* extracts, as compared to the untreated control group and the other extracts. In addition, extracts containing 80% *A. abdel-razek*, 80% *A. reticulata*, and volatile oil *A. glabra* exhibited a reduction in GSH activity compared to the control. Similarly, extracts containing 50% *A. reticulata* and 100% *A. muricata* showed a slight decrease in GSH activity compared to the other extracts and the control cells.

The HCT cell lines were exposed to various extracts and incubated for 48 h. It was observed that the volatile oil of *A. squamosa* and the 80% *A. reticulata* extracts led to a notable reduction in GSH levels. Conversely, the 80% *A. muricata* extract exhibited a stimulating impact on GSH levels. The Caco cell line was exposed to the most potent extracts, all of which resulted in a notable reduction in cytosolic GSH levels in the treated cells, except for the 80% *A. muricata* extract, which exhibited a rise in GSH levels.

#### 2.6.3. Wound Healing Assay In Vitro of Two Tumor Cells: Inhibition of Cell Migration by Selected Cytotoxic Extracts

During the process of epithelial–mesenchymal transition (EMT), cancerous cells of an epithelial nature undergo a transformation into highly motile and invasive cells with mesenchymal-like characteristics. This transformation ultimately leads to the emergence of disseminating tumor cells. It is worth noting that only a limited number of these disseminated cells are able to successfully metastasize. The presences of immune cells and inflammation in the microenvironment of the tumor have been identified as influential factors in driving the EMT process. However, there is a scarcity of studies that have explored the implications of EMT on tumor immunosurveillance. Our research not only demonstrates that EMT initiates metastasis, but also reveals that it renders the cancer cells more susceptible to the actions of natural killer (NK) cells. Furthermore, EMT contributes, to some extent, to the inefficiency observed in the metastatic process. Notably, when NK cells are depleted, spontaneous metastasis occurs while the growth of the primary tumor remains unaffected. The impact of selected extracts on the functional features of epithelial–mesenchymal transition (EMT) was assessed using a wound scratch assay, which involved examining migration and proliferation rates. In the wound scratch assay, it was observed that cells treated with the control vehicle migrated toward the wound but failed to close it after 24 h of incubation. In contrast, cells treated with the extracts (100 μg/mL) lost their ability to migrate after 48 h of treatment. In the experiment on wound healing, the movement of cells in response to the mechanical scratch wound was investigated. Figure 16 and Figure 17 depict images of the scratch regions taken at intervals of 0, 24, and 48 h. The elongated axis of the pipette tip used to create the scratch ought to be perpendicular to the bottom of the well and form a straight line in one direction. Two types of cell lines, namely the colon cancer cell line and the liver cancer cell line (HepG2), were employed in this study (Caco).

Firstly, Figure 16 illustrates the representative control for the Caco cell line at each time point, demonstrating that the wound was nearly half closed within 48 h. In order to assess the impact of different extracts of the volatile oil and the bioassay-guided differentiation process, the percentage of the open wound area after 48 h was determined. Our findings clearly indicate that the Caco cell line treated with 80% *A. muricata* and 50% *A. reticulata* extracts exhibited a significant inhibition of cell migration in comparison to the control at 0 h. Furthermore, the cell line treated with 100% *A. atemoya* and 50% *A. muricata* showed a slight effect on inhibiting cell migration capability. Caco cells treated with 100% *A. atemoya*, 50% *A. muricata*, and 80% *A. muricata* extracts impede cell motility and migratory capability in a 6-well plate after scratching the cell sheet and undergoing treatment for 48 h, whereas Caco cells treated with 50% *A. reticulata* extracts do not exhibit any effect on migration capability (Figure 16).

HepG2 cells were subjected to treatment with extracts of volatile oil *A. glabra* and various extract sources, including *A. glabra*, *A. abdel-razek*, *A. reticulata*, and *A. muricata*. These treatments resulted in the inhibition of cell motility, as evidenced by the impairment of migratory capability in a 6-well plate after scratching the cell sheet and a subsequent 24 h treatment. Notably, the 50% and 100% concentrations of *A. glabra* did not exert any influence on cell motility. Conversely, the 100% concentration of *A. reticulata* significantly inhibited cell migration, while the 50% concentration did not have an inhibitory effect nor when even using the 80% concentration. Furthermore, the treatment with 80% *A. abdel-razek* and 100% *A. muricata* yielded similar effects to *A. reticulata*, further supporting the observed inhibition of cell migration (Figure 17).

## 3. Discussion

### 3.1. Optimization of Extraction Method of Total Alkaloids

The extraction technique in this experiment resulted in the highest quantity of the extract obtained from the methanol extractions with concentrations of 100% and 80%, exceeding the minimum threshold of 50% set by the dichloromethane solvent. The analysis revealed that the yield of isolates was relatively lower compared to the one obtained by Gabriela Aguilar-Hernández et al. [70]. The outcome can be explained by the differences in the concentration of total alkaloids, which are directly linked to the variations in the percentage of alcohol utilized in the plants. The alkaloid is extracted from the organic solvent using aqueous solutions of 2% HCl acid, following the established methods. Subsequently, the acidified mixture is transferred to a separating funnel and subjected to extraction with dichloromethane until it attains a colorless appearance. The chloroform layer is discarded, and the acidified aqueous extract is filtered. Afterward, it undergoes conversion into alkali through the introduction of ammonia, and the pH level is adjusted to achieve a value of 11 (in Figure 2). The liquid alkali is extracted using chloroform. The chloroform layer that had been combined was dried through evaporation and subsequently weighed. The extraction approach corresponds to the findings of Ellaithy et al. [71].

### 3.2. Metabolomics and Fingerprint Profiles of Cultivated Annona Species

The histogram provides evidence that the observed statistic derived from the original data exhibited a comparable relationship between the SCoT and ISSR analysis for six species, namely *atemoya*, *reticulata*, *glabra*, *muricata*, *squamosa*, and *abdel-razek*. This outcome of the fingerprint analysis is also in concordance with the results obtained from the PCA statistical analyses, which indicate a partial similarity in the profiles of volatile compounds across different *Annona* species. The study on the metabolomics–metabolites and GC-MS analysis of volatile oils demonstrated a strong resemblance, with the following order: *Abdel Razek* > *squamosa*, *reticulata* > *muricata*, and *atemoya* > *glabra* (as depicted in Figure 8). The present investigation revealed that the highest concentrations of total alkaloids in various differentiation extracts were detected in 50% *A. squamosa*, 100% *A. reticulata*, 80% *A. squamosa*, 50% *A. glabra*, 50% *A. abdel-razek*, 80% *A. abdel-razek*, and 80% *A. muricata*, respectively, representing 0.3435%, 0.303%, 0.2018%, 0.2172%, 0.19200%, 0.11312%, and 0.1068% (total alkaloids)/25 g of dry aerial parts. These findings are consistent with the findings reported by El-Shemy and Hany [72]. The results obtained from the in vitro anticancer activities align with the fractions that are rich in total alkaloids, as illustrated in Figure 2 and Table 3.

The study successfully characterized 74 compounds identified from six *Annona* species using HRMS-UPLC. These compounds include 41 aporphine alkaloids, 9 oxoaporphines, one dioxoaporphine known as annonbraine, six benzylisoquinolines, one pallidine referred to as morphinandieone, one sarracine known as pyrrolidine, one nicotine alkaloid, one chlorantene B classified as sesquiterpenoids, and one protoberbrine identified as coreximine. The characterization was performed using LC/MSMS. An isoboldine dimethyl compound derivative (34) was observed in *A. salzmanii*. Paulo [73] identified three alkaloids in *A. cherimola*: isoboldine (27), magoflorine (24), and 7,4’-Di-O-methylcoclaurine (32). These alkaloids, namely aporphine alkaloids and benzylisoquinoline, were identified for the first time in our species.

The alkaloids of *Annona* sp. were analyzed using electrospray ionization (ESI) in only positive metabolite characteristics that were consistently present in all of the examined communities. A total of 43,547 signals were detected in the positive mode. Among them, 282 signals were matched in the first step, followed by 15,717 signals in the MS2 step. Finally, 4816 signals were proposed. The chromatogram displaying the most intense peak (base peak chromatogram or BPC) obtained using the indicated analytical methods is presented. Additionally, chromatograms at 280 nm, where isoquinoline alkaloids demonstrate absorption, and at 450 nm, where quaternary protoberberine alkaloids exhibit high activity, are also displayed. Isoquinoline alkaloids originate from the amino acid tyrosine following its conversion into 3,4-dihydroxyphenylethylamine (dopamine) and 4-hydroxyphenylacetaldehyde, which serve as intermediate molecules.

A group of vital secondary metabolites discovered in plants is referred to as aporphinoids. Within the realm of traditional medicine, numerous drugs have been employed over an extended period to address a range of problems, spanning from mild symptoms to more severe ailments. Over 500 aporphine alkaloids, derived from various plant groups, have been identified and found to possess potent cytotoxic characteristics. These alkaloids hold potential for the creation of anticancer medications [74].

The isoquinoline alkaloids originate from S-reticuline, a tetrahydrobenzyl isoquinoline composed of two tyrosine units forming its framework. S-(+)-norcoclaurine is synthesized through the condensation of these two components, facilitated by the enzyme S-norcoclaurine synthetase. N-methylcoclaurine is synthesized via the process of N-methylation and O-methylation occurring at position 6. Tetraoxygenated S-(+)-reticuline, a vital intermediate and fundamental component of aporphine alkaloids, is synthesized via the processes of 3 and 4 hydroxylations and 4-O-methylation. Aporphines are synthesized through the direct intramolecular oxidative coupling (ortho-ortho or ortho-para) of S-(+)-reticuline from the bisdienone radical state. The aporphines are generated through the substitution pattern of the tetrahydrobenzyl isoquinoline precursor. However, certain locations of O-substitution, such as C-3 or C-7, are formed by oxidizing the aporphinoid nucleus. S-adenosyl methionine has the ability to cause methylation at the C-7 site. Another method for producing aporphines involves the cyclization of an ortho-para-tetrahydroisoquinoline diradical, followed by direct protonation and a dienone-phenol rearrangement after a proaporphine intermediate [74].

The *Annona* sp. extracts and metabolites have been suggested to exert cytotoxic effects by interfering with the integrity of the mitochondrial membrane, leading to cell cycle arrest in the G0/G1 phase. Multiple signaling pathways that control metabolism, the promotion of metastasis, and the necrosis of cancer cells were found to hinder the induction of apoptosis [72]. The structural properties of alkaloids and derivatives of oxoaporphines are unique and may play a role in an anticancer process. Recent research has shown that this molecule has the ability to specifically attack cancer cells via many methods, such as producing reactive oxygen species, attaching to DNA, and inhibiting the telomerase enzyme [75,76].

### 3.3. Quantification of Malondialdehyde as a Biomarker of Oxidative Stress in Human Hepatoma HepG2 and T47D Cell Cultures

This study further highlighted the significance of hydro-alcoholic differentiation of *Annona* sp. in preventing lipid peroxidation. Lipids, particularly polyunsaturated fatty acids (PUFAs), can be easily damaged by reactive oxygen species (ROS) at the membrane. This results in the production of lipid hydro-peroxides and, subsequently, malondialdehyde (MDA), which is often employed as a biomarker to measure oxidative stress. Figure 14 displays the MDA levels for each treatment group. Significantly, there was a notable and statistically significant rise in MDA levels in HepG2 cells when treated with 50% and 100% *A. glabra*. This increase, as seen in Figure 14, reached about three-time higher levels compared to the negative control, which only included the medium.

Compared to the positive control, HepG2 cells treated with 80% *A. abdel-razek*, 80% *A. reticulata*, and 100% *A. muricata* showed a significant reduction (*p* < 0.05) in MDA levels. This suggests that the samples have the ability to protect the cells from lipid peroxidation caused by H_2_O_2_. In addition, the results of this study revealed that concentrations of 80% *A. abdel-razek* and 80% *A. reticulata* were sufficient to prevent lipid peroxidation. Increasing the concentration of these two extracts to 100% did not consistently result in a higher level of inhibition of lipid peroxidation. 

The T47D cells treated with a combination of 80% *A. glabra* and 80% *A. muricata* showed the least significant decrease in MDA levels compared to the negative control and other extracts. The analysis revealed that the extracts exhibited strong capabilities in scavenging hydroxyl radicals [77]. Moreover, the extracts derived from *Annona*, which constitute 80% of the composition, have shown the ability to protect liver cells from oxidative harm [78,79].

Prior studies have shown that pre-incubating HepG2 cells with aporphenoids and other derivatives can effectively inhibit lipid peroxidation [78]. Research utilizing aporphines, oxoaporphines, proaporphines, and other alkaloid-rich extracts from the *Annona* plant has demonstrated a decrease in lipid peroxidation in HepG2 cells. This emphasizes the crucial functions of antioxidant alkaloids in reducing oxidative harm. We agree with these findings [78]. This decrease may be attributed to the direct interactions between the evaluated stressors and the components of the medium, such as glucose, which could lead to a reduction in the generation of MDA. However, it should be noted that this possibility was not investigated. When the protein content was adjusted, the levels of MDA in the cytoplasm were the same as those seen in the control HepG2 cells (as shown in Figure 14 and Table 3) when these cells were directly exposed to t-BOOH or H_2_O_2_. These findings indicate that the stressors that were evaluated do not have a direct impact on the cytosol components. This could be due to the presence of antioxidant defense systems in the cytoplasmic contents, such as catalase and glutathione, which help mitigate their effects. Furthermore, these results indicate that the increased MDA levels observed in the cells treated with t-BOOH, as shown in Figure 14, were not influenced by substances present in the culture medium. Instead, they were likely caused by the processing of the cell lysates or by a direct impact of t-BOOH on the cell components, possibly through the peroxidation of membrane lipids.

### 3.4. The Frequency of Non-Enzymatic Antioxidants of GSH in Liver Cancer (HepG2) and Breast Cancer (T47D) as Tumor Grade Will Be Determined

The present work assessed the activity of the GSH antioxidant enzyme in two cell lines, HepG2 and T47D. Glutathione (GSH), a tripeptide, is the primary endogenous antioxidant synthesized by cells. It plays a crucial role in protecting cells from reactive oxygen species (ROS) such as free radicals and peroxides. The current consensus acknowledges that reactive oxygen species (ROS) and electrophilic compounds have the ability to cause DNA damage, while glutathione (GSH) can effectively counteract such harm. GSH can directly detoxify carcinogens by undergoing phase II metabolism and subsequently exporting toxic chemicals from the cell. The impact of antioxidants in 80% *A. atemoya* relative to the control was most pronounced in the T47D cell line, expressed as a percentage of total alkaloids (0.0919/25 g).

In HepG2, the most effective extract was found in 50% *A. reticulata*, with a concentration of 0.103/25 g. The presence of bioactive compounds in the plant can inhibit the harmful effects of reactive oxygen species (ROS) by modulating the function of antioxidant enzymes, hence preventing oxidative damage. Antioxidant enzymes play a crucial role in maintaining the redox balance of cells when they are exposed to oxidative stress. Antioxidant enzyme activity serves as a somewhat accurate indicator of oxidative stress and can also be utilized to predict plant responses to oxidative stress.

Since migration is a crucial stage in the spread of cancer, we used a wound healing assay to examine the impact of an extract from the *Annona* species that was chosen for its most active differentiation on the migration of HepG2 and Caco cells. After 48 h of treatment with an *Annona* species extract, both cells showed a dose-dependent suppression of migration. After 48 h of treatment and scratching the Caco cell sheet, surprisingly, the effect of 50% and 80% of the extracts of *A. muricata* on inhibition of migration was robust on highly metastatic Caco cells. HepG2 cell migration was markedly reduced by 50% of the *A. reticulate* extract.

## 4. Material and Methods

### 4.1. Collection of Plant Material and Isolation of DNA

Six *Annona* species trees were acquired from a privately-owned farm located in the Mansoriya district of Giza Zoo and Mazhar botanic garden area, under the Giza Governorate of Egypt. The trees were identified by Therese Labib, a specialized botanical consultant, and Dr. M. A. Gibali, from the Taxonomy Department at the Faculty of Science, Cairo University. The voucher specimens were stored at the Herbarium of the National Research Centre [18]. The aerial component (leaves and stem) was obtained from a private farm in Giza, Egypt, for the metabolomic analysis, while immature leaves were collected for DNA fingerprinting (Table 1). The DNA extraction was conducted using the DNeasy plant Mini Kit (QIAGEN). The DNA separation process was implemented using the methodology described by Dice [80]. The PCR procedure was conducted using the protocol described by Williams et al. [81]. The similarity matrices were generated utilizing the sophisticated Gel works ID program, namely the UVP-England Program. The genotypic associations were determined using dendrograms, which were constructed using the SPSS windows program (Version 10). DICE computer software was used to create the pair-wise difference matrix and phenogram among cultivars [21]. All of the utilized primers were able to amplify distinct and assessable bands. The unambiguous, replicable genetic variants amplified using the primers were recorded as 1 to indicate their presence or 0 to indicate their absence. The execution of all procedures adhered strictly to the applicable standards, regulations, and legislation. The authors obtained authorization from the National Research Centre Egypt to gather plant specimens from several locations. 

### 4.2. A Bioassay-Guided Differentiation Method Was Conducted for Six Annona Species and the Isolated Total Alkaloids

The methanol percentages of 50, 80, and 100 were used for the extraction of three different types of differentiation from six *Annona* sp. dry plants weighing 25 g. The extraction process was repeated three times using 100 mL of extracts each time. The gam-extracts were determined by measuring the weights in several species with varying alcohol concentrations. The total alkaloids were extracted by dissolving one gram of a crude extract from each differentiation step in HCl-acidified water at a pH of 3–4. This mixture was then subjected to three rounds of extraction with dichloromethane using a separating funnel. The water, which had been acidified, was converted to an alkaline state by introducing ammonia at a pH level of 11, and subsequently treated with dichloromethane once more. Dichloromethane was employed once as a sole extractant for the whole alkaloid fraction [12].

### 4.3. GC/MS Analysis (Determination and Identification)

The volatile oil content of the six aerial sections of *Annona* sp. harvested in September was determined. This was as per the aforementioned techniques [82]. In order to determine the proportion of volatile oil in each species, the recently trimmed aboveground parts that were considered waste were subjected to water distillation using a Clevenger apparatus. In addition, the volatile oil content of fresh aerial parts from six *Annona* species was determined through hydro-distillation [82]. The resulting essential oil was preserved in a deep freezer at −20 °C until the GC/MS analysis, following individual dehydration using anhydrous sodium sulphate. The average values of the oil content (%) were reported following the completion of the analysis in triplicate.

The constituents of *Annona* sp. essential oil were identified and analyzed using the gas chromatography/mass spectrometry (GC/MS) analysis, following the methodology described in our earlier study. The Central Laboratories of the National Research Centre in Cairo, Egypt, utilized an Agilent Technologies GC/MS system consisting of a gas chromatograph (7890B) and a mass spectrometer detector (5977A).

### 4.4. LC/MSMS Analysis


The dissolution of 10 mg from each of the eighteen extracts was performed in 1 milliliter of methanol, followed by filtration using a sarangi filter. Regarding the extract that had been diluted, dilution was repeated thrice.The LC/MS-ion-trap Esquire was equipped with ESI and operated in positive ion mode. The scan range was 100–3000 *m*/*z* and scan resolution was 13,000 *m*/*z*/s). A nebulizer, gas flow of 30 psi, 9.1 L/min, and temperature of 310 °C were used. The skimmer was set at -10.0 V [83].The LCMSMS system consisted of a Q-Exactive hybrid MS/MS quadrupole-Orbitrap mass spectrometer and UPLC (Waters, Milford, CT, USA). The separation of chromatographic components employed solvents (A:B) such as water acidified with 0.1% formic acid and acetonitrile, with a mobile phase flow rate of 0.3 mL/min. This was accomplished through a gradient program consisting of the following steps: from 0 to 15 min, the composition changed from 50% A to 50% B; from 15 to 22 min, the composition changed to 98% B and remained at this level for 22 min; from 22 to 23 min, the composition changed back to 95% A until 27 min, at which point the system returned to its initial conditions and was re-equilibrated for 3 min [83]. Our approach involved the processing of data obtained from mass spectrometric fragmentations of 74 alkaloid metabolites using ion mobility tandem mass spectrometry. This allowed us to generate a comprehensive fragmentation pattern, as well as retention time and MS/MS information.The MS-DIAL 4.60 tool, which utilizes the MSP format for the purpose of filtering noisy spectra through a fundamental spectral similarity computation, offers enhanced and standardized untargeted metabolomics by exporting the four portions of the imported raw MS data to a common output format (abf). The identification of the distinct compounds involved the utilization of precise molecular masses (within a range of less than 5 ppm), mass spectra, retention periods, internet databases (ChEBI, Metlin, PubChem, and KNApSAck, ChemSpider), as well as literature data [84,85].


### 4.5. Model of Antiproliferation for Six Annona sp.

#### 4.5.1. Estimation of Potential Cytotoxicity of Extracts on Cell Lines Using Sulphorhodamine-B (SRB) Assay

All chemicals and reagents for the cell culture and bioassays were purchased from Lonza (Verviers, Belgium) or Sigma-Aldrich (Steinheim, Germany). The method was carried out according to that of [86]. The sensitivity of different cell lines (HepG2, Caco, HCT, MCF7, and T47D; ATCC, USA) was taken from the Vacsera (Giza, Egypt) [87] to the extracts determined using the SRB assay. SRB is a bright pink amino-xanthene dye with two sulfonic groups. It is a protein stain that binds to the amino groups of intracellular protein under mild acidic conditions to provide a sensitive index of cellular protein content. Then, further assays, such as a cell cycle analysis, were performed to investigate the anticancer effect.

The percentage of cell survival was calculated as follows: Survival fraction = O.D. (treated cells)/O.D. (control cells).

The IC_50_ values (the amounts of TAM and 3-BP required to inhibit 50% cell growth) were obtained using dose-response curve-fitting models (GraphPad Prism software, version 5). Each experiment was run three times.

##### Estimation of Total Lipid Peroxide Content (Measured as Malonaldialdehyde)

Malonaldialdehyde (MDA) was used according to the method described in [88]. The lipid peroxidation products were estimated in MCF7 and T47D cells through determination of thiobarbituric acid reactive substances (TBARS) that were measured as MDA. The latter is a decomposition product of the process of lipid peroxidation and is used as an indicator of this process. The principle depends on colorimetric determination of the pink pigment product, resulting from the reaction of one molecule of MDA with two molecules of thiobarbituric acid (TBA) at low pH (2–3) and at a temperature of 95 °C for 45 min. The resulting color product was measured at 535 nm.

##### Estimation of Reduced Glutathione Content

Reduced glutathione was determined according to [89]. The process is based on the sulfhydryl, SH, group in glutathione (GSH) reducing the thiol reagent, 5,5-dithiobis (2-nitrobenzoic acid) (DTNB, Ellman’s reagent), to generate the yellow chromophore, 5-thionitrobenzoic acid (TNB), which is detected spectrophotometrically at 405 nm. Precipitation of protein thiols by trichloroacetic acid (TCA) is carried out before the addition of Ellman’s reagent. Serial dilutions of GSH were used to set up a standard curve.

#### 4.5.2. Wound Healing Assay

The scratch assay method was employed to evaluate the migration rates of HepG2 and Caco cells. A cell density of “2 × 105 cells” was introduced into each well of a 24-well plate and incubated with a complete medium under conditions of 37 °C and 5% CO_2_. Following 24 h of incubation, the monolayer confluent cells were horizontally scraped using a sterile P200 pipette tip. The debris was subsequently eliminated through a thorough washing with PBS. The cells were subjected to samples with a concentration of 100 μg/mL. In order to serve as a negative control, cells without any treatment were employed. The scratch, which represented the wound, was captured at 0 h using phase contrast microscopy at ×40 magnification before the incubation with the samples. After 24 h of incubation, a second set of images were taken. In order to ascertain the migration rate, the images were analyzed using “image J” software Version 1.54, and the percentage of the closed area was measured and compared with the initial value obtained at 0 h. An increase in the percentage of the closed area indicated the migration of cells. The experiments conducted yielded these results [90].
$$\mathrm{Wound}\;\mathrm{closure}\;(\%)=\frac{\left(Measurement\;at\;0\;h-Measurement\;at\;24\;h\right)\times 100}{Measurement\;at\;0\;h}$$

## 5. Conclusions

The study of metabolomics, which focuses on the analysis of metabolites, and the examination of the constituents of volatile oils using GC-MS demonstrated a significant resemblance. Specifically, the following patterns were observed: *abdel-razek* > *squamosa*, *reticulata* > *muricata*, and *atemoya > glabra*. The utilization of liquid chromatography in conjunction with high-resolution tandem mass spectrometry can serve as a powerful method for the identification of several alkaloids and metabolites, spanning across different classes, in *Annona* sp. These data can be utilized to evaluate the efficacy of this plant for prospective medicinal applications and understand the mechanisms of alkaloid production in 24 preparations. All alkaloids and their derivatives exhibited clear antioxidant, anticancer, and antiproliferative effects in our investigation. Plant extracts and phytochemicals have been closely linked to cytotoxicity, encompassing metastasis and necrosis of cancer cells, as well as the breakdown of the mitochondrial membrane. This disruption prevents cells from advancing into the G0/G1 phase and triggers cell death. The literature assessment of the current work indicates that pre-incubating HepG2 cells with aporphenoids and other derivatives can reduce lipid peroxidation. Research using aporphines, oxoaporphines, proaporphines, and other *Annona* extracts abundant in alkaloids revealed a decrease in lipid peroxidation within HepG2 cells. This highlights the noteworthy contribution of antioxidant alkaloids in safeguarding against oxidative harm. The migration of HepG2 and Caco cells was most significantly impacted by the extract that was the most active. In particular, the extracts of *A. muricata*, at concentrations of 50% and 80%, exhibited a suppression of migration that was dependent on the dose (Figure 18). This suppression was observed after 48 h of treatment and the creation of scratches on the Caco cell sheet. Notably, this effect was particularly strong on Caco cells that were highly metastatic. Furthermore, the extract of *A. reticulata*, at a concentration of 50%, demonstrated the greatest reduction in migration of HepG2 cells.

## Figures and Tables

**Figure 1 pharmaceuticals-17-00103-f001:**
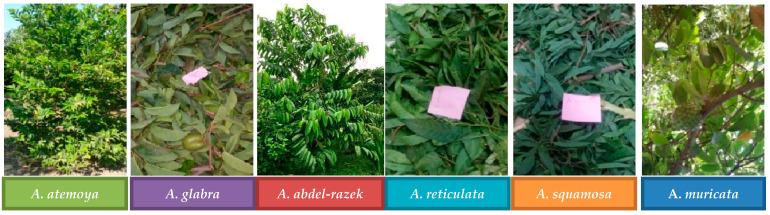
Six genotypes of *Annona* sp.

**Figure 2 pharmaceuticals-17-00103-f002:**
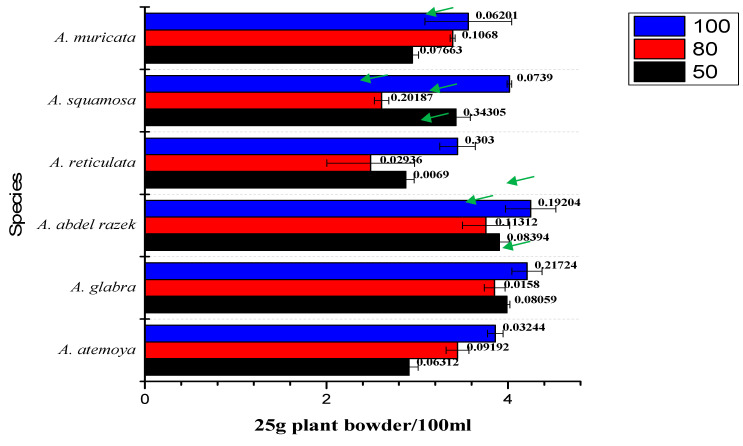
Bioassay-guided differentiation of different 50, 80, and 100 methanolic extracts and the number superscript beside columns means the percentage of total alkaloids g/25 g of plants in each differentiation. Green arrow that indicates a high concentration.

**Figure 3 pharmaceuticals-17-00103-f003:**
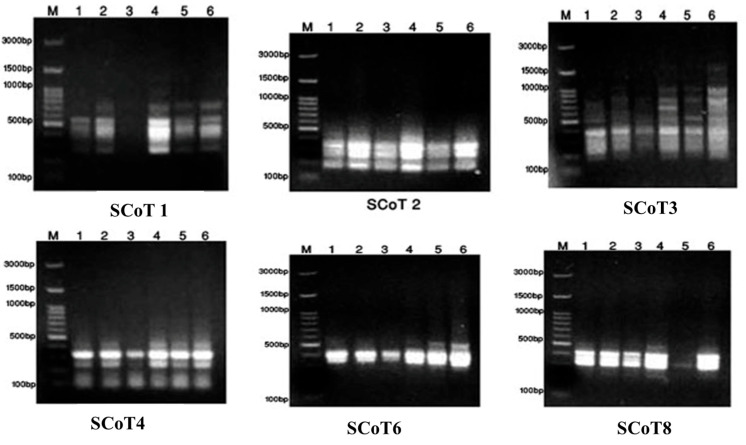
Display of the DNA of primers in the six studied *Annona* samples with SCoT. As 1 = *A. atemoya*, 2 =*A. glabra*, 3 = *A. abdel-razek*, 4 = *A. reticulata*, and 5 = *A. squamosa*, then 6 = *A. muricata*.

**Figure 4 pharmaceuticals-17-00103-f004:**
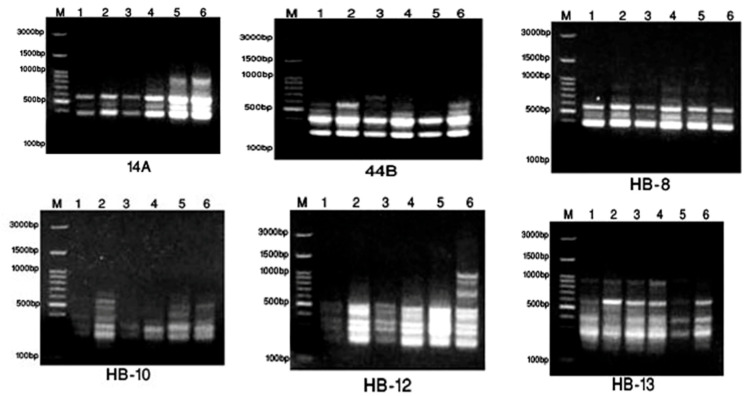
Display of the DNA of primers in the six studied *Annona* samples with ISSR. As 1 = *A. atemoya*, 2 = *A. glabra*, 3 = *A. abdel-razek*, 4 = *A. reticulata*, and 5 = *A. squamosa*, then 6 = *A. muricata*.

**Figure 5 pharmaceuticals-17-00103-f005:**
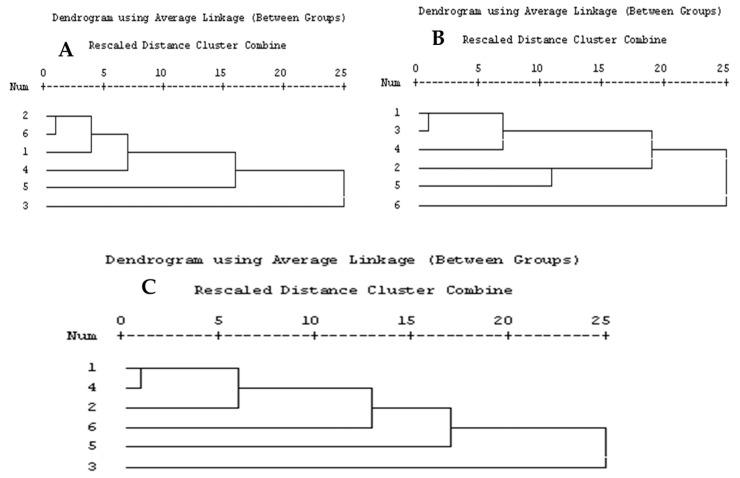
(**A**) Dendrogram of SCoT analysis, (**B**) dendrogram of ISSR analysis, and (**C**) dendrogram of combination SCoT and ISSR analysis for six *Annona* sp.

**Figure 6 pharmaceuticals-17-00103-f006:**
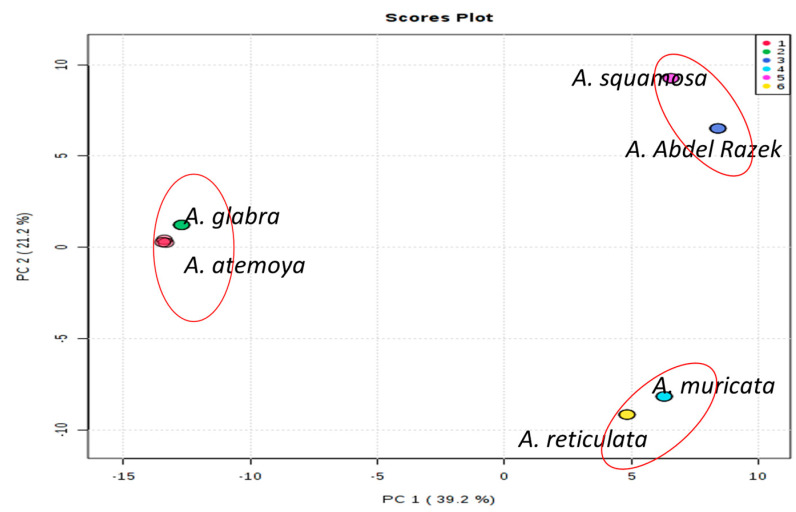
PCA statistical analysis showing similarity of volatile compound profiles between *Annona* species. These results are in semi-agreement with ISSR and SCoT data but typical agreement with metabolomics profiles. A red circle means similarity between the two species.

**Figure 7 pharmaceuticals-17-00103-f007:**
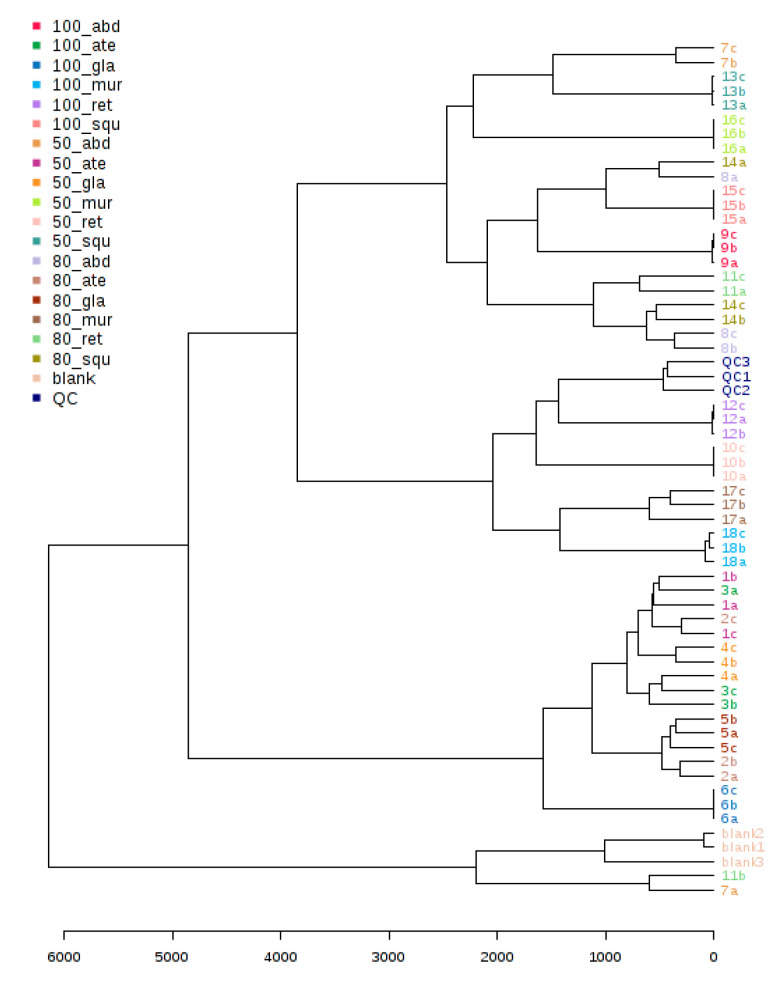
Illustration in positive mode of the similarity of the previous results between *abdel-razek* > *squamosa*, *reticulata* > *muricata*, and *atemoya > glabra.* The cross-validated QC and different colors of dots indicate different groups of metabolites.

**Figure 8 pharmaceuticals-17-00103-f008:**
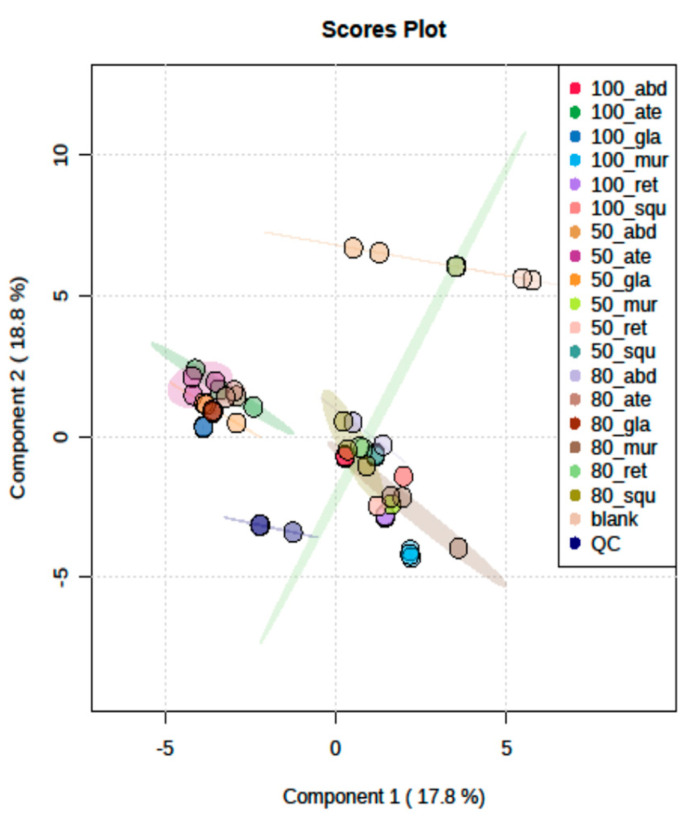
Illustration of the PLS-DA loading plot of different extract species.

**Figure 9 pharmaceuticals-17-00103-f009:**
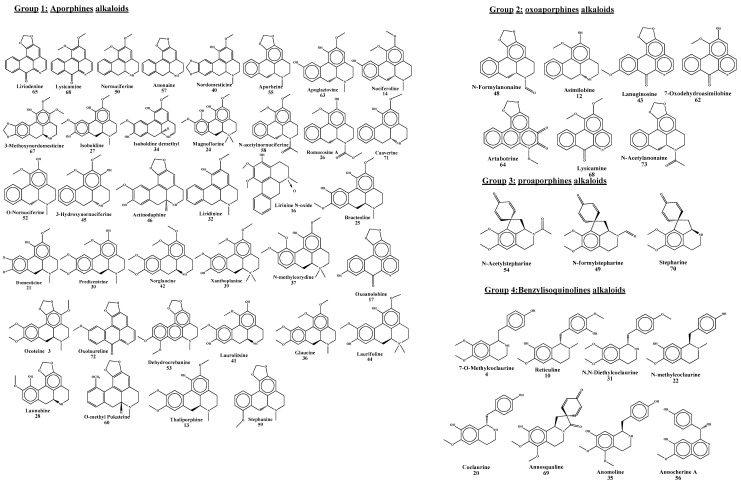
Chemical structures of the alkaloids present in 18 extracts of six *Annona* and characterized by the HRMS-UPLC platforms. Compounds are numbered according to Table 4.

**Figure 10 pharmaceuticals-17-00103-f010:**
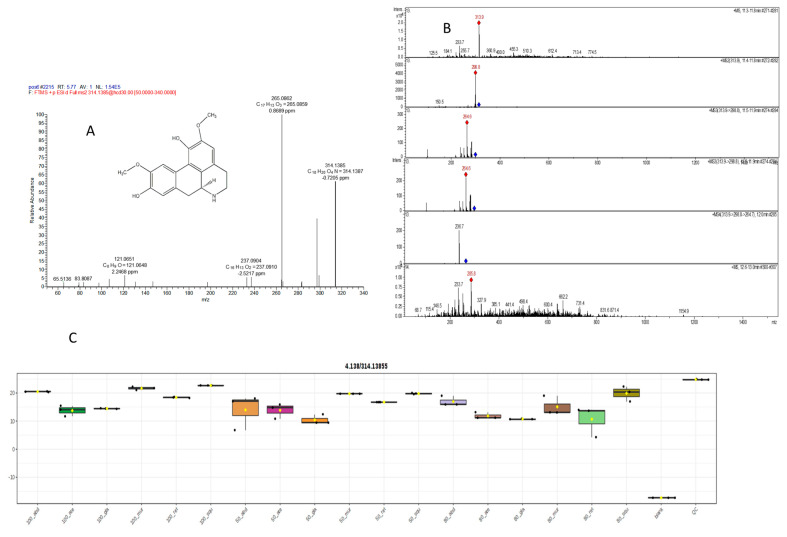
(**A**) LC/MSMS of compound (34) norisoboldinedemethyl, (**B**) different observed fragmentation in low abundance, and (**C**) the different concentrations in different alcohol percentages of extracts in six species.

**Figure 11 pharmaceuticals-17-00103-f011:**
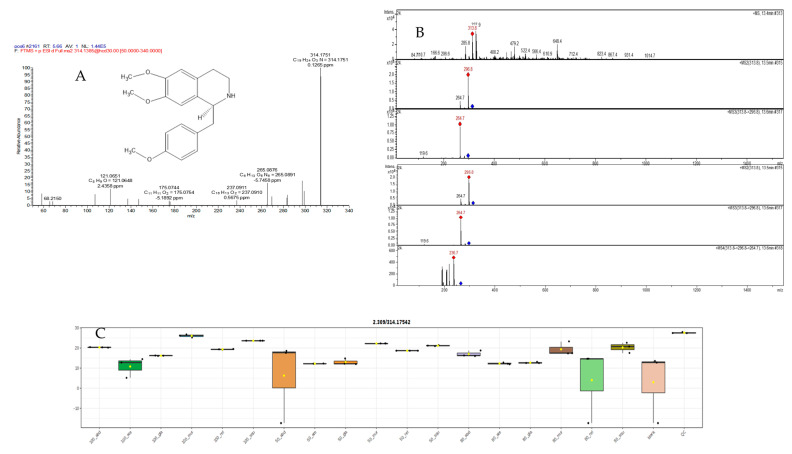
(**A**,**B**) LC/MSMS of compound (31) N,N-dimethylcoclaurie and (**C**) the different concentrations in different alcohol percentages of extracts in six species.

**Figure 12 pharmaceuticals-17-00103-f012:**
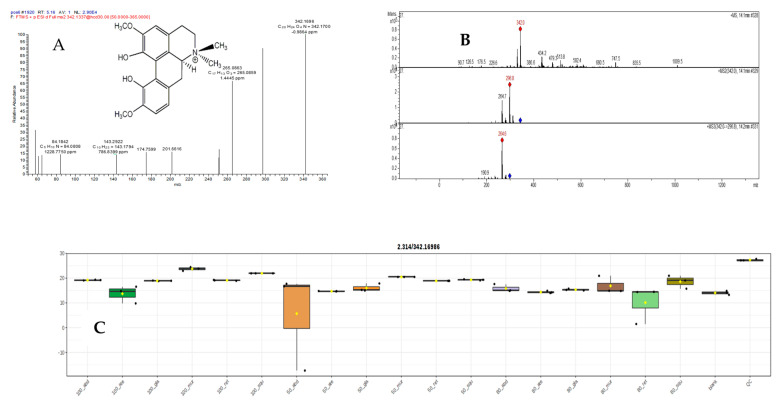
(**A**,**B**) LC/MSMS of compound (24) magoflorine and (**C**) the different concentrations in different alcohol percentages of extracts in six species.

**Figure 13 pharmaceuticals-17-00103-f013:**
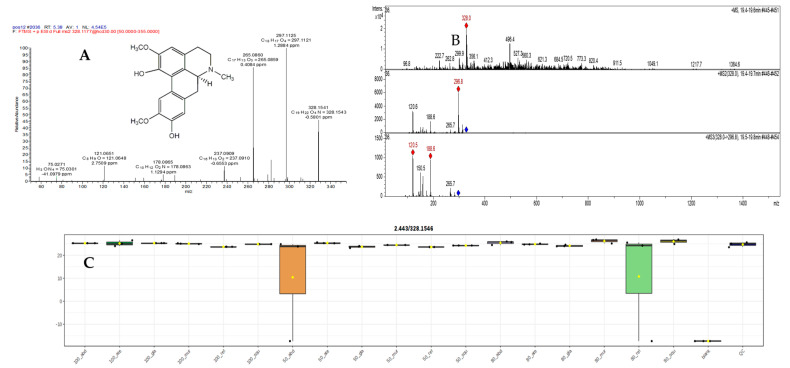
(**A**,**B**) LC/MSMS of compound (27) isoboldine and (**C**) the different concentrations in different alcohol percentages of extracts in six species.

**Figure 14 pharmaceuticals-17-00103-f014:**
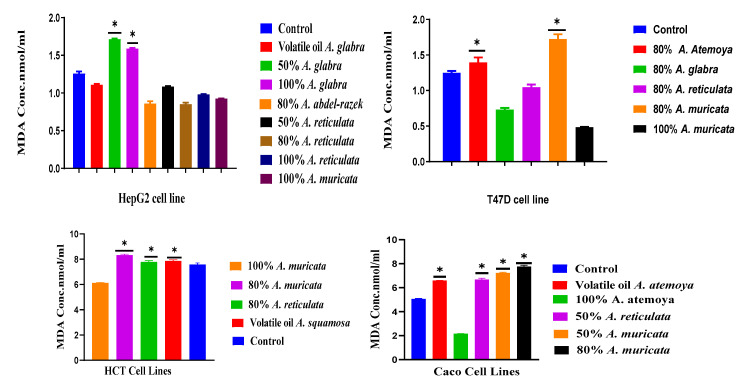
HepG2, T47D, HCT, and Caco cells were exposed to various extracts through the utilization of malondialdehyde. *: means high-treated extracts.

**Figure 15 pharmaceuticals-17-00103-f015:**
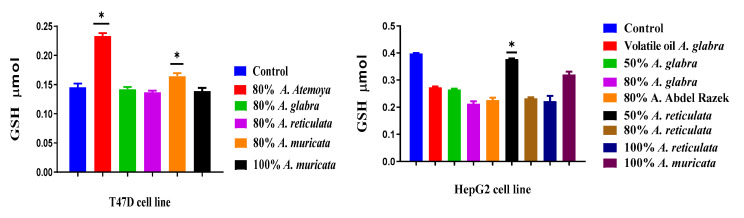
T47D, HepG2, HCT, and Caco cells were subjected to various extracts through the utilization of reduced glutathione. *****: means high-treated extracts.

**Figure 16 pharmaceuticals-17-00103-f016:**
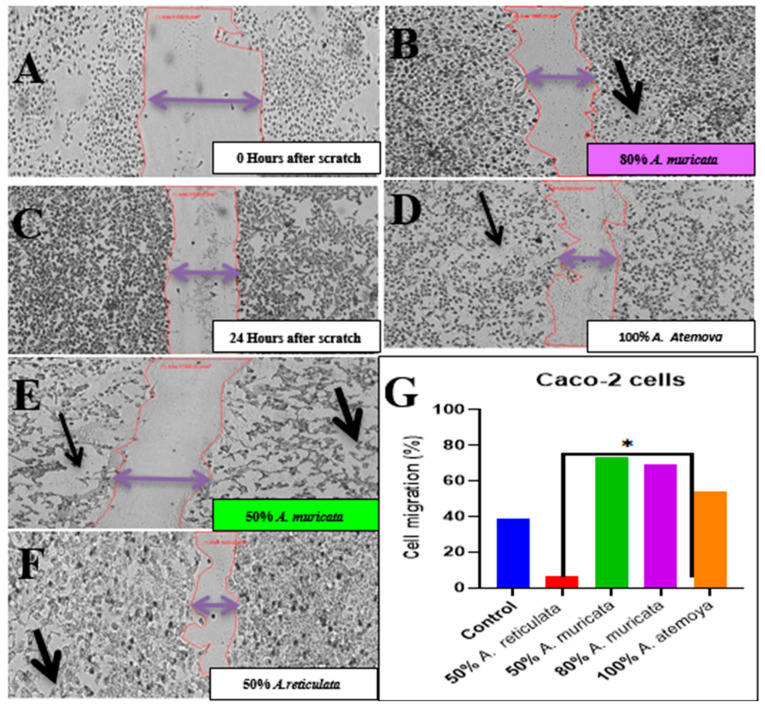
(**A**,**B**) A total of 0 and 24 h after scratch control groups. (**C**–**F**) The extract from *Annona* species differentiation effectively suppresses cellular movement by inhibiting both migratory capabilities and motility. This inhibition is observed after creating a scratch on the cellular sheet and subjecting it to a 24 h treatment. Detailed examination of the obtained gap contour overlays reveals that there are groups of cells moving faster than the neighbor lattice for several cell lines. Black arrow indicates dead cells and stoppage of migration cells. (**G**) The quantitative wound cell migration was employed to evaluate this phenomenon. *****: means high-treated extracts.

**Figure 17 pharmaceuticals-17-00103-f017:**
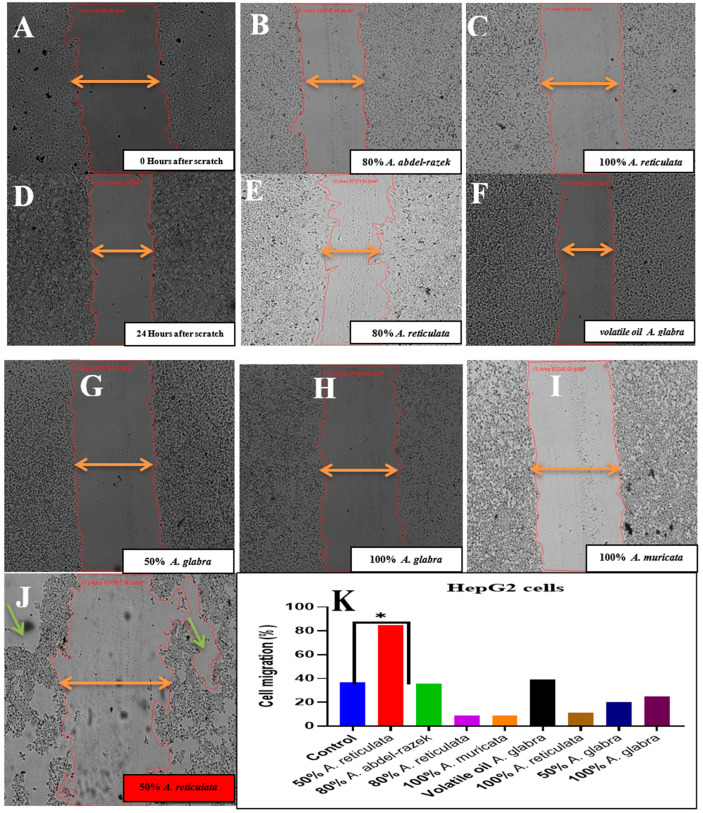
(**A**,**B**) A total of 0 and 24 h after scratch control groups. (**C**–**J**) The extract from Annona species differentiation effectively suppresses cellular movement by inhibiting both migratory capabilities and motility. This inhibition is observed after creating a scratch on the cellular sheet and subjecting it to a 24 h treatment. This event occurs after 48 h of scratch ((**J**), green arrows). Such cells start to detach from the cell lattice and may re-enter back. It was observed more frequently for HepG2 (enlargement). Detailed examination of the obtained gap contour overlays reveals that there are groups of cells moving faster than the neighbor lattice for several cell lines. (**K**) The quantitative wound cell migration is employed to evaluate this phenomenon. *: means high-treated extracts.

**Figure 18 pharmaceuticals-17-00103-f018:**
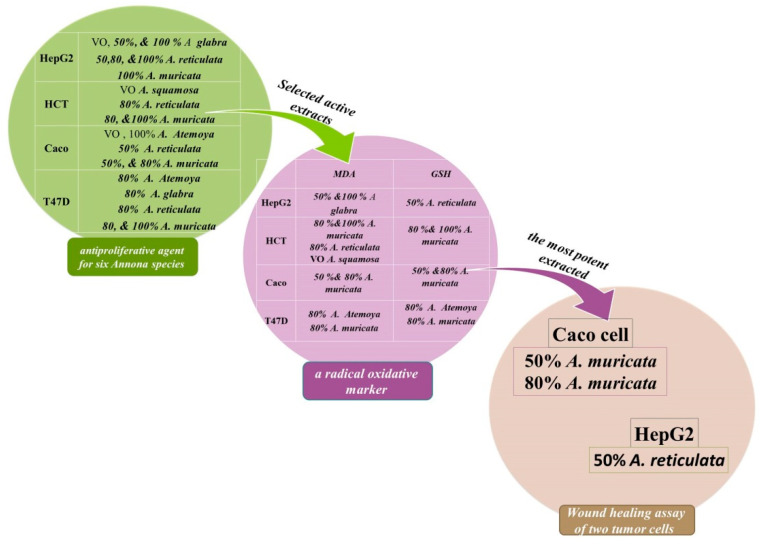
These data can be utilized to evaluate the efficacy of this plant for prospective medicinal applications and understand the mechanisms of alkaloid production in 24 preparations. All alkaloids and their derivatives exhibited clear antioxidant, anticancer, and antiproliferative effects in our investigation. Plant extracts and phytochemicals have been closely linked to cytotoxicity, encompassing metastasis and necrosis of cancer cells, as well as the breakdown of the mitochondrial membrane [49,54]. The three most promising extracts were finalized as follows: 50% *A. muricata*, 80% *A. muricata*, and 50% *A. reticulate* [91].

**Table 1 pharmaceuticals-17-00103-t001:** List of nucleotide sequences of ISSR and SCoT procedure.

No.	Primer	ISSR Sequence	Primer	SCoT Sequence
1	A-14	5′ CTC-TCT-CTC-TCT-CTC-TTG 3′	SCoT 1	5′ ACG-ACA-TGG-CGA-CCA-CGC 3′
2	B-44	5′ CTC-TCT-CTC-TCT-CTC-TGC 3′	SCoT 2	5′ ACC-ATG-GCT-ACC-ACC-GGC 3′
3	HB-8	5′ GAG-AGA-GAG-AGA-GG 3′	SCoT 3	5′ ACG-ACA-TGG-CGA-CCC-ACA 3′
4	HB-10	5′ GAG-AGA-GAG-AGA-CC 3′	SCoT 4	5′ ACC-ATG-GCT-ACC-ACC-GCA 3′
5	HB-12	5′ CAC-CAC-CAC-GC 3′	SCoT 6	5′ CAA-TGG-CTA-CCA-CTA-CAG 3′
6	HB-13	5′ GAG-GAG-GAG-C 3′	SCoT 8	5′ ACA-ATG GCT-ACC-ACT-ACC 3′

**Table 2 pharmaceuticals-17-00103-t002:** The main constituents of the essential oil of *Annona* sp.

Peak No.	RT	Identified Constituents	Chemical Formula	MS (M/e)				% Components in *Annona* sp.					
				*m*/*z*	No. of Scans	Main Significant Fragments	Base Peak	*A. atemoya*	*A. glabra*	*A. abdel-razek*	*A. reticulata*	*A. squamosa*	*A. muricata*
1	11.333	α-Thujene	C_10_H_16_	136	17	121, 105, 93, 77, 65, 53	93	0	0	0	1.89	0.2	0
2	11.641	α-Pinene	C_10_H_16_	136	17	121, 93, 77, 67	93	1.31	4.85	8.03	1.85	4.94	9.1
3	12.178	Camphene	C_10_H_16_	136	17	121, 107, 93, 79	93	0.35	1.57	0.35	0.19	1.91	0.16
4	12.959	β-Pinene	C_10_H_16_	136	19	121, 107, 93, 69, 53	93	0	0.4	8.43	1.47	1.62	9.45
5	13.088	Sabinene	C_10_H_16_	136	19	121, 105, 93, 77, 69, 53	93	0	0	0.27	11.9	0.78	2
6	13.694	β-Myrcene	C_10_H_16_	136	16	121, 93, 69, 41	93	3.06	6.36	1.11	1.49	0.27	3.17
7	14.201	α-Phellandrene	C_10_H_16_	136.	16	121, 93, 77	93	2.28	5.96		0.41	0	0.13
8	14.906	β-Cymene	C_10_H_14_	134	17	119, 103, 91, 77, 51	119	1.04	1.08	0	0.29	0	0
9	15.034	Eucalyptol	C_10_H_18_O	154	17	139, 121, 108, 93, 81, 71, 55	81	0	0	0	0	0	2.41
10	15.099	D-Limonene	C_10_H_16_	136	17	121, 107, 93, 79, 68, 53	93, 68	2.42	3.41	3.27	1.45	3.25	3.08
11	15.792	β-Ocimene	C_10_H_16_	136	16	105, 93, 79, 53	93	2.95	8.61	2.52	0.39	0.24	0.7
12	16.171	γ-Terpinene	C_10_H_16_	136	18	121, 105, 93, 77, 65	93	0	0	0	1.68	0.18	0.15
13	16.526	Iso-β-terpineol	C_10_H_18_O	154	20	121, 93, 71, 55	71	0	0	0	0.13	0	0
14	17.261	α-Terpinolene	C_10_H_16_	136	16	121, 105, 93, 79, 67, 53	93	0	0	0	0.75		0.18
15	17.704	Linalool	C_10_H_18_O	154	21	136, 121, 93, 71, 55	71	0.4	0.51	0.72	0.52	0.2	0.33
16	18.514	Trans-p-2-Menthen-1-ol	C_10_H_18_O	154	21	139, 111, 93, 79, 55, 43	43	0	0	0	0.17	0	0
17	20.828	Trans-3(10)-Caren-2-ol	C_10_H_16_O	152	20	137, 119, 109.000, 95, 81, 69, 41	109	0	0.27	0	0	0	0
18	20.548	Terpinen-4-ol	C_10_H_18_O	154	18	125, 93, 71, 55	71	0	0	0	3.08	0.41	0.24
19	20.956	α-Terpineol	C_10_H_18_O	154	18	121, 107, 93, 81, 59, 43, 31	59	0	0.23	0	0.19	0	0.15
20	21.592	Acetic acid, octyl ester	C_10_H_20_O_2_	172	17	152, 112, 70, 43	43	0.56	0	0	0	0	0
21	22.285	Cis-3-Hexenyl isovalerate	C_11_H_20_O_2_	184	16	152, 103, 67, 57	67	0	0	0	0	0	0.11
22	24.226	Bornyl acetate	C_12_H_20_O_2_	172	17	154, 121, 108, 95, 80, 55	95	0.35	0.52	0	0	0.9	0.13
23	25.963	δ-Elemene	C_15_H_24_	204	16	189, 161, 136, 121, 93, 77, 55, 41	121	0	0	21.4	4.33	5.22	6.99
24	26.33	α-Cubebene	C_15_H_24_	204	16	189, 161, 133, 105, 91, 81, 55	105	0	0	0	0.21	0.27	0.25
25	27.222	α-Copaene	C_15_H_24_	204	16	189, 161, 133, 119, 93, 81, 55	161, 119	1.52	0.89	0.58	1.88	1.04	3.06
26	27.414	γ-Muurolene	C_15_H_24_	204	20	190, 161, 147.1100, 133, 105, 91, 69, 55	161	0	0.48	0	0	0	0
27	27.484	Bicyclo [5.2.0]nonane, 4-methylene-2,8,8-trimethyl-2-vinyl-	C_15_H_24_	204	25	189, 161, 133, 105, 93, 81, 67, 53	81	0	0	0	1.42	0	1.21
28	27.718	β-elemene	C_15_H_24_	204	24	189, 175, 161, 147, 119, 107, 93, 67, 55	161, 93	2.4	1.75	8.9	11.21	4.96	9.72
29	28.009	(+)-α-Himachalene	C_15_H_24_	204	20	189, 175, 161, 119, 93, 55	93	0	0.4	0	0	0	0
30	28.027	α-Gurjunene	C_15_H_24_	204	13	189, 178, 161, 147, 133, 119, 105, 79, 55	105	0	0	0	0.15	1.43	0.15
31	28.242	Cis-Caryophyllene	C15H24	204	28	189, 165, 117, 132, 119, 119, 91, 55	105	0.33	4.1	0.88	0	10.77	1.4
32	28.749	Caryophyllene	C_15_H_24_	204	29	189, 161, 133, 120, 105, 93, 79, 69, 55	133	28.09	19.59	5.37	11.66	16.63	6.22
33	28.936	Β-cubebene	C_15_H_24_	204	27	189, 161, 133, 105, 79	161	0.91	0.43	1.05	1.21	0.55	0.86
34	29.047	Trans-α-Bergamotene	C_15_H_24_	204	26	189, 161, 133, 119, 107, 93	119, 93	4.2	2.39	5.15	0.83	0	0.33
35	29.425	7-epi-α-Cadinene	C_15_H_24_	204	29	189, 161, 147, 133, 119, 105, 91, 79, 55	161, 91	0.36	0	0.4	0.37	0	0
36	29.659	Cis-β-Farnesene	C_15_H_24_	204	16	161, 147, 133, 105, 93, 69	69	0.45	0.2	0	0.17	0	0
37	29.787	Humulene	C_15_H_24_	204	22	175, 147, 121, 107, 93, 80	93	5.22	3.84	1.49	2.46	5	1.75
38	30.545	γ-Muurolene	C_15_H_24_	204	17	189, 161, 147, 133, 105, 79, 55	161	0.3	0.56	0.46	1.58	2.08	1
39	30.795	Germacrene D	C_15_H_24_	204	26	177, 161, 147, 133, 119, 105, 79, 55	161	9.5	4.98	13.06	10.74	4.47	6.32
40	30.993	β-Selinene	C_15_H_24_	204	19	189, 161, 147, 133, 105, 79	105, 161	0	0	0.28	0	0.26	0.51
41	31.081	δ-Selinene	C_15_H_24_	204	20	189, 175, 161, 147, 133, 91, 55	189, 161	0	0	0.29	0	0	11.12
42	31.355	γ-Elemene	C_15_H_24_	204	35	161, 121, 93, 41	121, 93	0	0	2.3	0	2.79	0
43	31.366	(+)-γ-Gurjunene	C_15_H_24_	204	20	189, 179, 161, 133, 121, 93, 79, 67	121, 93	0	0	0	6.95	0	0
44	31.413	α-Muurolene	C_15_H_24_	204	22	189, 161, 133, 119, 105, 91, 79, 55	105	0.96	0.33	0	0	0.77	0
45	31.578	α-Farnesene	C_15_H_24_	204	16	189, 161, 133, 107, 93, 69, 55	93	0.82	0.38	0	0	0	0
46	31.675	β-Bisabolene	C_15_H_24_	204	20	189, 161, 134, 107, 93, 79, 69	93, 69	1.38	0.73	0	0	0	0.27
47	32.165	Cubedol	C_15_H_26_O	224	29	207, 189, 161, 145, 91, 79, 67.0700, 55	161	0	0	0	0	0.43	0
48	32.008	Trans-γ-cadinene	C_15_H_24_	204	22	191.1200, 161.1000, 133.1000, 105.1000, 79.0900	161	0.36	0.64	0.34	0.34	2.27	0.96
49	31.669	β-Bisabolene	C_15_H_24_	204	25	189, 161, 134.1200, 105, 93, 79, 69	161	0	0	0	0.93	0	0
50	32.147	Cubedol	C_15_H_26_O	224	33	207, 189, 161, 135, 79, 55	161	0	0	0	0.53	0.38	0.34
51	34.514	(-)-Globulol	C_15_H_26_O	224	33	204, 189, 177.1200, 161, 122, 105, 81, 55	43	0	0	0	0	0	0.14
52	32.346	δ-Cadinene	C_15_H_24_	204	24	189, 161, 119, 91, 69	161	2.19	2.17	0.73	1.51	2.48	1.81
53	32.678	α-Patchoulene (1.alpha.,3a.alpha.,7.alpha.,8a.beta.)-	C_15_H_24_	204	24	189, 161, 119, 93, 79	93	1.84	1.03	0	0.16	0	0
54	33.628	Elemol	C_15_H_26_O	224	39	189, 161, 135, 107, 79, 59	93	0	0	2.45	0.33	0.67	0
55	34.153	Nerolidol	C_15_H_26_O	222	32	189, 161, 136, 107, 93, 69, 55	69	1.03	1.11	0.34	0.75	0.53	0.29
56	34.823	Squalene	C_30_H_5_0	381	38	207, 161, 95, 81, 69, 53	69	0	0	0	0	0	0.6
57	35.138	(−)-Spathulenol	C_15_H_24_O	220	37	220, 205, 187, 159, 119, 105, 91, 79, 43	43	4.65	0.35	0	0.58	0.84	0.53
58	35.365	Caryophyllene oxide	C_15_H_24_O	220	36	205, 161, 135, 121, 109, 79, 69, 43	43	4.15	3.68	0.28	1.19	1.77	0
59	35.767	Veridiflorol	C_15_H_26_O	222	31	204, 177, 149.1100, 135, 121, 107, 81, 43	43	0.4	0.47	0	0.5	0.89	0
60	36.624	β-Ionone	C_13_H_20_O	192	22	205, 177, 161, 121, 91, 55	177	0	0	0	0	0.28	0
61	36.233	β-Eudesmol	C_15_H_26_O	222	222	204, 177, 164, 149, 133, 107, 81, 59	149	0	0	0	0	0	0.27
62	37.09	Germacrene D-4-ol	C_15_H_26_O	222	20	207, 161, 93	93	0	0	3.71	0.54	0	0
63	37.265	β-Guaiene	C_15_H_24_	204	35	179, 145, 119, 105, 79, 55	119	0	0	0.8	0	0.87	0
64	37.318	Cubenol	C_15_H_26_O	222	30	204, 179, 161, 119, 105	119	0.49	1.22	0.37	0.9	0	0.28
65	37.428	Iso-spathulenol	C_15_H_24_O	220	36	204, 177, 162, 133, 119, 105, 91, 79, 55	119, 91	0	0	1.39	0	0.77	0
66	37.941	Tau-Cadinol	C_15_H_26_O	222	50	204, 189, 161, 121, 95	161	3.33	0.52	1.52	2.19	8.27	3.45
67	38.116	Torreyol	C_15_H_26_O	222	22	204, 189, 161,136, 119, 79	161	0.57	3.7	0	1.64	0.7	0.13
68	38.244	Cis-α-Bisabolene	C_15_H_24_	204	38	136, 93	93	1.26	0.48	0	0	0	0
69	38.361	Tau-Muurolol	C_15_H_26_O	222	37	204, 161, 134, 105, 81	161	0	0	0.64	0.25	0.38	0
70	38.495	α-Cadinol	C_15_H_26_O	222	28	204, 189, 161, 137, 121, 95, 81, 55	95	3.66	0.96	0.94	0	3.83	2.61
71	40.716	Alloaromadendrene oxide-(1)	C_15_H_24_O	220	23	177, 135, 107, 69	69	0.34	1.94	0	0	0	0
72	40.926	Bergamotol, Z-.alpha.-trans-	C_15_H_24_O	220	77	187, 119, 93	93	1.69	1.23	0	0	0	0
73	43.024	Farnesol, acetate	C_17_H_28_O_2_	253	22	189, 136, 93, 69	69	0.45	0.7	0	0	0	0.37
74	48.293	Neoisolongifolene, 8-bromo-	C_15_H_23_Br	282	40	225, 203, 175, 91, 41	203	0.32	1	0.23	0.39	2.56	0
75	49.208	Chlorpyrifos	C_9_H_11_C_l3_NO_3_PS	313	24	313, 257, 196, 170, 124, 96	96	0	0	0	0.12	0	0
76	50.146	Geranyllinalool	C_20_H_34_O	313	50	203, 135, 69	69	0.41	0.43	0.54	0	0.91	0.67
		Total Identification						98.3	96.45	100.59	95.87	99.97	95.1
		Un-identification compound						2.02	3.55	−0.59	4.13	0.03	4.9
		Oxygenated compounds						22.48	17.84	13.7	13.61	23.03	13.05
		Non-oxygenated compounds						75.82	78.61	86.89	82.26	76.94	82.05

**Table 3 pharmaceuticals-17-00103-t003:** IC_50_ of five cell lines as antiproliferative agent for six *Annona* species.

Different Extract	Volatile Oils of *Annona* sp.						Different Alcoholic Extracts of *Annona* sp.																	
	*A. Atemoya*	*A.* *glabra*	*A.* *abdel-razek*	*A. * *reticulata*	*A. squamosa*	*A. * *muricata*	*A. Atemoya*			*A. glabra*			*A. abdel-razek*			*A. reticulata*			*A. squamosa*			*A. muricata*		
IC_50_ (μg/mL)							1	2	3	4	5	6	7	8	9	10	11	12	13	14	15	16	17	18
							50%	80%	100%	50%	80%	100%	50%	80%	100%	50%	80%	100%	50%	80%	100%	50%	80%	100%
**HepG2** **Liver Cell**	32	** 18 **	>100	>100	50	>100	80	36.5	43	** 14.5 **	77	** 13 **	>100	** 10 **	41	** 12.5 **	** 10 **	** 13.5 **	56	50	>100	29	31	** 10.5 **
**MCF7** **Breast Cell**	77	26.5	81.5	>100	42.5	>100	8.5	85	9	19	42.5	8.5	99	28	83.5	36.5	29.5	10.5	11	>100	>100	10	10.5	9.5
**HCT** **Colon Cell**	73	>100	>100	>100	** 35.5 **	54	>100	>100	>100	>100	>100	>100	>100	>100	>100	>100	** 48 **	68	>100	>100	>100	>100	** 36.5 **	** 24 **
**Caco** **Colon Cell**	** 22 **	>100	99	>100	49.5	33	44	49.5	** 24.5 **	93	>100	50	>100	>100	>100	** 12 **	47	85.5	>100	>100	>100	** 21 **	** 15 **	68
**T47D** **Breast Cell**	>100	>100	>100	>100	>100	>100	>100	** 18.5 **	63.5	>100	** 23 **	>100	>100	>100	>100	>100	** 40.5 **	53	>100	>100	>100	37.5	** 38 **	** 43 **

Six volatile oil samples as *A. atemoya*, *A. glabra*, *A*. *abdel-razek*, *A. reticulata*, *A. squamosa*, and *A. muricata*; then, eighteen varying concentrations of alcoholic extract (50, 80, 100%), respectively. IC_50_ above 100 μg/mL was considered using GraphPad Prism analysis.

**Table 4 pharmaceuticals-17-00103-t004:** Alkaloids characterized in *Annona* sp. using HRMS-UPLC.

No.	RT	Tentative Identification	Chemical Formula	High-Resolution MS Data		∆ppm	*Annona* sp.																		ID 1, 2, and 3	Ref.
				[M + H]^+^ *m*/*z* Measured and Calculated	Major Fragments		*A. Atemoya*			*A. glabra*			*A. abdel-razek*			*A. reticulata*			*A. aquamosa*			*A. muricata*				
							1	2	3	4	5	6	7	8	9	10	11	12	13	14	15	16	17	18		
**Group 1: Aporphine**																										
3	2.21	Ocoteine ^C^ Thalicmine	C_21_H_23_NO_4_	370.1720, 370.1721	208.1178, 185.9153, 148.7571, 109.2159, 74.0610	−0.0224		*																	1	[24]
6	2.28	Apomorphine ^B^	C_17_H_17_O_2_N	268.1326, 268.1040	251.1066, 219.0805, 191.0805, 136.0620, 97.0295	−4.303																		*	2	[25]
12	3.39	Asimilobine ^A^	C_17_H_17_O_2_N	268.1341, 268.1333	251.1067, 236.0836, 219.0806, 191.0856, 163.0761, 163.0761	3.3151						*													1	[26]
13	3.85	Thaliporphine ^B^	C_20_H_24_NO_4_^+^	342.1703, 342.1700	297.1121, 265.0858, 237.0905, 178.0861, 123.0440, 58.0661	0.7974																			1	[27]
14	4.08	Nuciferoline ^C^	C_19_H_21_NO_3_	312.1592, 312.1594	259.0969, 265.0862, 205.4610, 171.2698, 144.4292, 104.0781, 76.6994	0.6723			*																1	[28]
16	4.34	Lirinine N-oxide ^C^	C_19_H_21_O_4_N	328.1544, 328.1543	297.1124, 283.0966, 265.0868, 178.0864, 116.0532	0.709						*													-	[28]
19	4.81	N-acetyl-3-methoxynornantenine ^C^	C_22_H_23_NO_6_	398.1598, 398.1598	377.8857, 298.1073, 176.0712, 155.0394, 65.0490	−0.1366																		*	-	[28]
21	5.02	Domesticine ^C^	C_19_H_19_NO_4_	326.1385, 326.1387	295.0963, 265.0859, 244.4052, 225.6086, 128.8098, 118.4798, 78.7120, 66.3822	0.694								*											1	[29]
24	5.16	Magnoflorine ^B^	C_20_H_24_NO_4_^+^	342.1696, 342.1700	297.1120, 265.0863, 251.0703, 201.6616, 174.7599, 143.2922, 84.1842	−0.9864						*													1	[30]
25	5.19	Bracteoline ^C^	C_19_H_21_NO_4_	328.1542, 328.1543	297.1128, 265.0860, 237.0910, 178.0865, 121.0652, 75.0269	0.3941																			1	[31]
26	5.24	Romucosine A ^B^	C_19_ H_19_O_4_N	326.1389, 326.1387	295.0965, 265.0859, 237.0911, 193.3112, 183.8072	0.5224			*	*	*										*				1	[32]
27	5.38	Isoboldine ^B^	C_19_H_21_NO_4_	328.1541, 238.1543	297.1125, 285.1139, 265.0860, 237.060, 237.0909, 178.0865, 121.0651, 58.0660	−0.5801												*							1	[33]
28	5.48	Launobine ^C^	C_18_H_17_O_4_ N	312.1228, 312.1230	297.1003, 263.1003, 263.0699, 242.381, 64.7326	−0.6449								*											1	[34]
30	5.62	Predicentrine ^B^	C_20_H_24_NO_4_^+^	342.1701, 342.1700	311.1278, 296.1034, 279.1015, 264.0779, 248.0827, 178.0862, 58.0660	0.3515													*						1	[35]
32	5.69	Liridinine ^C^	C_19_H_21_NO_3_	312.1593, 321.1594	280.0738, 263.0703, 235.0760, 205.0648, 118.0079	−0.379			*																-	[28]
34	5.77	Derivative of isoboldinedemethyl ^C^	C_18_H_19_NO_4_	314.1385, 314.1387	283.1325, 265.0862, 237.0904, 197.1719, 147.051, 121.061	−0.7207						*													-	[36]
36	5.8	Glaucine ^A^ O,O-dimethylisoboldine	C_21_H_26_NO_4_	356.1856, 356.1856	325.1433, 311.1281, 294.1255, 279.1017, 237.0919, 213.5202, 93.0936	0.0754												*							1	[26]
37	5.8	N-methylcorydine ^B^	C_21_H_26_NO_4_	356.1850, 356.1856	311.1278, 279.1016, 264.0788, 164.0421, 147.0438, 85.2029	−1.8747								*											1	[37]
38	5.85	Nuciferine ^B^	C_19_H_21_O_2_N	296.1646, 296.1645	278.1177, 251.1067, 219.0807, 145.4770, 109.4571, 88.9068, 58.0662	0.3591									*										1	[33]
39	5.86	Xanthoplanine ^B^	C_21_H_26_NO_4_	356.1859, 356.1856	311.1275, 279.1022, 264.5139, 206.1176, 186.3449, 137.7469, 70.8474	0.867															*				1	[37]
40	5.90	Nordomesticine ^A^	C_18_H_17_ O_4_N	312.1226, 312.1230	294.1129, 259.5082, 159.2854, 106.766	−1.3293																			1	[37]
41	5.96	Laurolitsine ^B^	C_18_H_19_NO_4_	314.1383, 314.1387	297.1125, 265.0860, 237.0921, 147.0439, 121.0653, 92.6976	−1.1091						*													1	[38]
42	5.98	Norglaucine ^A^	C_20_H_24_NO_4_^+^	342.1695, 342.1700	311.1279, 296.1046, 279.1016, 265.0851, 248.032, 149.1094, 75.1861	−1.5215																			1	[39]
43	6.03	Asimilobine ^A^	C_17_H_17_NO_2_	368.1331, 268.1332	251.1066, 219.805, 191.0854, 116.0090	-0.5546									*										1	[26]
46	6.57	(-)-Actinodaphine ^C^	C_18_H_17_O_4_N	312.1235, 312.1230	295.0964, 265.0857, 237.0913, 200.9195, 177.3906, 158.8117, 106.9259	1.4084																			3	[40]
47	6.77	Xylopine ^A^	C_18_H_17_O_3_N	296.1284, 296.1281	265.0861, 188.0704, 125.3992, 90.0052	0.9034									*										1	[28]
44	6.07	Laurifoline ^C^	C_20_H_24_NO_4_^+^	342.1705, 342.1700	311.1267, 297.1129, 265.0868, 178.0860, 64.9947, 58.0660	1.4217								*											1	[38]
48	6.87	N-formylanonaine ^A^ (-)-N-formylanonaine	C_18_H_15_O_3_N	294.1124, 294.1125	263.0702, 236.1060, 127.6403, 114.6490	−0.2504			*						*										1	[41]
50	6.94	Nornuciferine ^A^Sanjoinineia Daechualkaloid E	C_18_H_19_O_2_N	282.1480, 282.1489	265.1222, 250.0989, 243.1039, 164.0425, 129.0027	−2.8872														*					1	[41]
52	7.17	N-methylasimilobine ^B^ O-nornuciferine	C_18_H_19_NO_2_	282.1247, 282.1237	265.1223, 250.0986, 234.1042, 186.6200, 118.7022, 806066	3.4064																			1	[42]
55	7.33	Roemerine ^B^ (-)-Aporheine	C_18_H_17_NO_2_	280.1340, 280.1332	249.0910, 234.1497, 207.0805, 150.0269, 117.1255, 77.8964	2.7374																*			1	[43]
57	7.91	Anonaine ^A^	C_17_H_15_O_2_N	266.1175, 266.1176	249.0910, 219.0805, 191.0854, 174.0443	−0.2353			*			*													1	[41]
58	7.96	N-acetylnornuciferine ^C^	C_20_H_21_NO_3_	324.1596, 324.1594	309.1001, 171.3005, 111.1272	0.4823																			1	[44]
59	8	Stephanine ^B^	C_19_H_19_NO_3_	310.1438, 310.1438	279.1014, 249.010, 194.0215, 75.1965	0.093									*							*			1	[45]
60	8.06	O-methyl pukateine ^C^	C_19_H_20_NO_3_	310.1426, 310.1438	279.1014, 249.0904, 85.4723 62.2940	−3.6462								*											3	[28]
61	8.85	N-formylstepharine ^C^	C19H19NO4	326.1377, 326.1387	309.1119, 279.1015, 266.0927, 129.4605, 102.2371, 91.6374	−3.0333			*																1	[28]
63	9	Apoglaziovine ^A^	C_18_H_19_NO_3_	298.1442, 298.1438	281.1166, 249.0910, 221.1166, 194.1585, 127.0428, 64.0436	1.325																*			1	[28]
67	10.88	3-Methoxynordomesticine ^C^	C_19_H_19_O_5_N	342.1336, 342.1336	292.8240, 279.1023, 265.0850, 136.3306, 93.0376	0.004		*								*				*					1	[28]
71	12	Caaverine ^B^	C_17_H_17_O_2_N	268.1312, 268.1332	235.0067, 148.0763, 131.0494, 121.0652, 107.0496, 59.0501	0.5546							*												1	[46]
**Group 2: Oxoaporphine alkaloids**																										
1	1.64	Annonbraine ^A^	C_19_ H_11_O_4_N	318.0385, 318.0397	290.0431, 242.0245, 218.9947, 137.8421, 90.9476	−1.219					*	*												*	-	[41]
17	4.44	Oxoanolobine ^B^	C_17_H_19_NO_4_	302.1479, 302.1387	253.0852, 225.0901, 191.0707, 159.0440, 121.0653, 107.0497, 69.7030	−2.6683															*				1	[47]
53	7.24	Dehydrocrebanine ^C^	C_20_H_19_NO_4_	338.1385, 338.1387	323.1148, 177.0544, 91.6380, 61.0048	−0.4889			*																1	[48]
62	8.67	7-Oxodehydroasimilobine ^B^	C_17_H_11_NO_3_	278.0813, 278.0812	263.0581, 236.9391, 213.9227, 149.0237, 105.1089	0.3544			*																1	[49]
64	9.58	Artabotrine ^B^	C_18_H_11_O5N	322.0708, 322.0710	278.40501, 164.5077, 157.0504, 110.9065, 71.1918	−0.7101									*										1	[50]
65	10.03	Liriodenine ^A^	C_17_H_9_O_3_N	276.0654, 276.0655	249.9080, 226.9840, 208.8835, 127.9462, 110.9124, 64.1885	−0.4367				*															1	[51]
66	10.3	Lanuginosine ^A^	C_18_H_11_NO_4_	306.0760, 306.0761	274.2528, 230.1798, 166.1937, 140.2873, 70.0659	−0.412									*										1	[42]
68	10.9	Lysicamine ^A^ (Oxonuciferine)	C_18_H_13_NO_3_	292.0965, 292.0968	277.0733, 248.0707, 218.1884, 163.0764, 128.8807, 81.0039	−1.0021												*						*	1	[41]
72	12.84	Oxolaureline ^C^ 10-Methoxyliriodenine	C_18_H_10_NO_4_	306.0764, 306.0761	267.3079, 240.9942, 131.1421, 102.4868, 78.4019	0.9839															*				2	[28]
73	15.13	N-acetylbongaridine ^C^ (-)-N-acetylanonaine	C_19_H_17_O_3_N	308.1284, 308.1281	249.0909, 219.0807, 156.2953, 134.3902, 86.0606	0.8682															*				1	[52]
**Group 3: Proaporphine alkaloids**																										
49	6.88	(-)-N-formylstepharine ^C^	C_19_H_19_NO_4_	326.1382, 326.1387	296.0966, 265.0860, 237.0910, 171.9669, 69.8721	−1.4426																			1	[53]
54	7.27	N-acetylstepharine ^C^	C_20_H_21_NO_4_	340.1540, 340.1543	328.6469, 297.4160, 218.9519, 160.7969, 129.9211, 92.8186, 71.1164	−1.0083																			1	[54]
70	11.7	Stepharine ^C^	C_18_H_19_NO_3_	298.1447, 298.1438	221.9773, 177.0549, 145.0285, 105.0706, 69.1952	3.0651												*							1	[41]
**Group 4: Benzylisoquinoline and isoquinoline alkaloids**																										
4	2.22	7-O-methylcoclaurine ^C^	C_18_H_21_NO_3_	300.1595, 300.1594	283.1328, 269.1170, 223.1120, 192.1016, 176.0754, 137.0598, 121.0651, 107.0496, 58.0661	0.2159						*													1	[30]
10	2.35	Reticuline ^A^	C_19_H_23_O_4_N	330.1697, 330.1700	192.1019, 175.0754, 137.0597	0.08374	*					*						*							1	[55]
20	5.01	Coclaurine ^B^	C_17_H_19_NO_3_	300.1594, 300.1594	283.1328, 269.1171, 237.0916, 175.0754, 137.0597, 121.0652, 89.0605	0.0126						*													1	[30]
22	5.1	N-methylcoclaurine ^B^	C_18_H_21_O_3_N	300.1594, 300.1541	283.1328, 269.1171, 237.0916, 175.0754, 137.0597, 121.0652, 89.005	0.0126						*						*							1	[56]
31	5.66	N,N-dimethylcoclaurine ^C^	C_19_H_24_NO_3_^+^	314.1751, 314.1751	297.1111, 265.0876, 237.0911, 175.0744, 147.0446, 121.0651	0.1265						*													-	[36]
56	7.49	Annocherine A ^A^	C_17_H_15_NO_4_	298.1076, 298.1074	280.0967, 192.0660, 168.9584, 129.7724, 60.6217	0.6373																			1	[57]
35	5.78	Anomoline ^B^	C_18_H_21_NO_4_	316.1540, 316.1543	267.1016, 239.1066, 191.0703, 159.0440, 121.0652	-0.9883																			1	[58]
69	11.03	Annosqualine ^A^	C_19_H_19_NO_5_	342.1337, 342.1336,	324.1225, 265.0876, 222.0770, 189.0428, 84.9932	0.2873															*				1	[59]
**Other compounds**																										
5	2.33	Pallidine ^B^ Morphinandienone	C_19_H_21_NO_4_	328.1536, 328.1543	247.1118, 256.0858, 237.0908, 203.5709	−2.2541																			1	[60]
7	2.28	Nicotinamide 2 Nicotine alkaloid	C_6_H_6_N_2_O	123.055	96.0555, 80.0498, 67.0552	−0.1804																		*		[61]
8	2.28	Chlorantene B ^C^ Sesquiterpenoid alkaloids	C_15_H_19_O_5_N	294.1323, 294.1336	276.1443, 230.1388, 212.1283, 173.9608, 132.1021, 97.0292, 86.0972	0.1276			*		*	*													1	[62]
2	1.99	6-(Alpha-D-glucosaminyl)-1D-myo-inositol	C_12_H_24_O_10_N	342.1402, 342.1403	306.1182, 283.1075, 282.1075, 240.0867, 204.0874, 162.0761, 127.0393	0.3943	*													*						-
9	2.37	Isoleucine ^C^ Amino acids	C_6_H_12_NO_2_	132.1021, 132.1019	114.0553, 97.0289, 86.0972, 68.0504, 58.0661	1.1019						*													1	[61]
11	2.39	Sarracine ^C^ Pylrrolidine	C_18_H_27_O_5_N	338.1972, 338.1975	285.1080, 254.1135, 237.0869, 211.1084, 159.0766, 114.0553	1.001	*																			[63]
15	4.29	Phenylacetaldehyde	C_8_H_8_O	121.0651, 121.0647	103.0547, 93.0706, 53.0397	−0.3087																*			-	-
18	4.45	Coreximine ^A^ Protoberberine	C_19_H_21_O_4_N	328.1545, 328.1543	313.1313, 265.0865, 235.8464, 192.1019, 178.0864, 151.0754, 117.0734, 93.0376, 58.0660	0.4428																			1	[64]
23	5.15	Piperolactam C ^B^ Aristolactamalkaloids	C_18_H_15_O_4_N	310.1073, 310.1074	279.0889, 165.6389, 101.8022, 70.9469, 63.6019	−0.2731		*										*							1	[65]
29	5.55	Scopoletin ^C^	C_10_H_9_O_4_	193.0498, 193.0495	165.0547, 133.0285, 147.0444	1.3956																			1	[66]
33	5.73	Unknown	C_20_H_31_O_6_N	382.2227, 382.2224	273.1493, 191.0694, 161.5542, 132.7013, 65.1538	0.8729								*											-	-
45	6.36	3-Hydroxynornuciferine ^B^	C_18_H_19_NO_3_	298.1434, 298.1438	281.1170, 249.0911, 221.0973, 160.1834, 94.6420, 59.5692	−1.234																*			1	[67]
51	6.96	Unknown	C_22_H_17_ON_2_	326.1411, 326.1414	309.1119, 294.0886, 278.0939, 265.0858, 251.1050	−0.9582																			-	-
74	16.27	Benzophenone-3 ^C^	C_14_ H_12_O_3_	229.0856, 229.0859	151.0391, 105.0340, 54.08002	−0.7861																		*	2	[68]

As Letter ^A^: from *Annona* species, ^B^: *Anonnaceae* family, ^C^: first time recorded in Annonaceae family, then ID 1 is from KNApSAK and 2 from ChemSpider and 3 from pubchem. * indicates identified compound from this species and this differentiation (1–18) of extract as 50, 80, and 100 percentage for each species, respectively.

## Data Availability

The datasets generated and/or analyzed during the current study are available in the (https://www.dropbox.com/scl/fo/4m7hyq7wlhoap9t84t7yj/h?dl=0&rlkey=6lbhlk1im6ansr9x97ztv1zn) repository.

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
