# Peer review of "Comprehensive Tools of Alkaloid/Volatile Compounds–Metabolomics and DNA Profiles: Bioassay-Role-Guided Differentiation Process of Six *Annona* sp. Grown in Egypt as Anticancer Therapy"

_pharmaceuticals, 2024, doi:10.3390/ph17010103_

Round 1

Reviewer 1 Report

Comments and Suggestions for Authors

The presented study, titled "Comprehensive Tools for Metabolomics and DNA Profiling: Bioassay-Guided Differentiation of Six Annona Species Grown in Egypt for Anti-Cancer Therapy," delves into the metabolomic analysis of six Annona tree species renowned for their bioactive compounds with anti-cancer, antioxidant, and anti-migratory properties in tropical and subtropical regions. DNA fingerprinting using SCoT and ISSR primers revealed polymorphism levels of 45.16% and 35.29%, respectively, among the varieties. The metabolomic analysis employed GC-MS to compare volatile oil profiles in the six Annona species. Additionally, HPLC-ESI-MSn and UPLC-HESI-MS/MS were utilized for the characterization of 74 alkaloid compounds in a bioassay-guided differentiation process, evaluating their antiproliferative activities on five cell lines. High-throughput LC/MS facilitated in-depth investigations into secondary metabolite changes during the bioassay-guided differentiation process. Notably, in vitro testing on cell cultures demonstrated concentration-dependent cytotoxic effects on all cell lines (HepG2, HCT, Caco, Mcf-7, and T47D) treated with six volatile oil samples for 48 hours. To enhance the impact and utility of the manuscript, the following revisions are suggested:

-About Annona Species, provide additional information on the chemical constituents responsible for the reported medicinal benefits. 

-Figure 8 “Chemical structures of the alkaloids”, clarify the 2D structures in a more explicit manner. 

- In the section discussing MDA levels, offer a clearer interpretation of the results. Explain the significance of changes in MDA levels and their relation to the antioxidant properties of Annona species.

- Consider reorganizing the table 2 for better clarity, or alternatively, include it in the supplementary materials.

Comments on the Quality of English Language

Moderate editing of English language required

Author Response

Dear Reviwer #1:

We highly appreciate your constructive comments which we are sure will add value to our manuscript. We have prepared a point-by-point response hoping that they will be satisfactory. Your comments are in red, and our response is in black.

Regards…

We considered all comments and modifications with thorough revisions and language editing made in the manuscript.

#Find the attachment file.

Reviewer 2 Report

Comments and Suggestions for Authors

1.      What is the study's novelty? The phytochemicals and biological activities of Annona species have already been documented. How does this manuscript distinguish itself from the articles listed below?

A.      KAZMAN, Bassam SM Al; HARNETT, Joanna E.; HANRAHAN, Jane R. The phytochemical constituents and pharmacological activities of Annona atemoya: a systematic review. Pharmaceuticals, 2020, 13.10: 269.

B.      AMUDHA, P.; VARADHARAJ, VANITHA. Phytochemical and pharmacological potential of annona species: a review. Asian J Pharm Clin Res, 2017, 10.7: 68-75.

C.      CHOWDHURY, Shadman Sakib, et al. Screening of antidiabetic and antioxidant potential along with phytochemicals of Annona genus: A review. Future Journal of Pharmaceutical Sciences, 2021, 7: 1-10.

D.      NUGRAHA, Ari Satia, et al. Anti-infective and anti-cancer properties of the Annona species: Their ethnomedicinal uses, alkaloid diversity, and pharmacological activities. Molecules, 2019, 24.23: 4419.

E.       LEITE, Débora OD, et al. Annona genus: traditional uses, phytochemistry and biological activities. Current Pharmaceutical Design, 2020, 26.33: 4056-4091.

2.      2. Italics are not used for in vitro and botanical names. It should be italicized throughout the manuscript.

3.      Linguistic and typographical errors make this manuscript difficult to read. It need extensive modification and should be revised by English editing services.

Few examples are given blow.

1.      Line 36: “exanimate”; Line 176 “Annona sp .extracts.”

2.      Line  “The medicinal importance of the Annona species trees are due to the presence of some special secondary metabolites like alkaloids asisoquinoline, aporphine, proaporphine, and oxoaporphine groups [12], glycosides, terpens, cyclopeptides, flavonoids, resins, volatile oils, tannins and acetognins etc. 80 [13] Annona reticulata observations to be used as a chemopreventive agent in cancer therapy according to Furthermore, A. muricata is a popular medicinal remedy in Africa, America, and India for the treatment of cancer [14].”

3.      Line 118-120: “Fig (1) the three different successive extraction (50, 80 &100) methanol percentage of six Annona sp. 25g dry plant extracted by 100 ml extracts three times with shaking 170rpm. Its gives different weights inside species as appeared.”

4.      Line 115: “per-formed”

4.      The Fond used in this manuscript differs across lines.

5.      Line No. 111 “The current study included aerial parts of six different genotypes of Annona sp., in- 111 cluding A. atemoya; A. glabra; A. muricata; A. squamosa and A. Abdel Razek that were gathered in the Giza governorate of Egypt in Fig (17).” Mentioned Six species but given only Five species names. In addition, Fig. 17, only figures are visible but no species names in the figure.

6.      There is no information on which samples were collected for metabolite profiling or sample preparations for LC/MS and GC/MS.

7.      Why is the extract concentration different in each cell line throughout biological activities? Explain.

8.      Fig. 8 must be reorganized because numerous structures overlapped.

9.      Figure legends are not informative.

10.  Poorly written results and discussion, as well as linguistic issues, make it difficult to read.

11.  Overall, this paper has to be extensively edited in order to fulfill the journal's standards. 

Comments on the Quality of English Language

Linguistic, typo, punctuation marks errors makes this MS difficult to read. 

Author Response

Dear Reviewer, #2: The comments were highly insightful and enabled us to greatly improve the quality of our manuscript the following pages are our point -by-point responses to each of the comments.

Find the attached file

Reviewer 3 Report

Comments and Suggestions for Authors

Dear Author,

Thank you for submitting your research article entitled "Comprehensive Tools of Alkaloid/Volatile Compounds-Metabolomics & DNA Profiles: Role-Bioassay-Guided Differentiation Process of Six Annona sp. Grown in Egypt as Anti-Cancer Therapy" to our journal. I have carefully reviewed the manuscript and would like to provide you with comprehensive feedback and suggestions for improvement.

1. The main focus of this study is the utilization of metabolomics and DNA profiling to conduct biological identification and classification of six Annona species. This approach, which combines GC-MS-based metabolomics and DNA fingerprinting, addresses the limitations of using either method alone. By employing both techniques, you have effectively laid the groundwork for the future clinical applications of Annona. This research has the potential to evaluate the therapeutic uses of Annona and shed light on the biosynthesis of alkaloids in the 24 extracts, making it highly valuable for further investigation.

2. In the Results section, it would be beneficial to provide additional information regarding the rationale behind using three different concentrations of methanol (50%, 80%, and 100%) for alkaloid extraction from Annona. Clarifying the source of the data or explaining the reasoning behind these specific concentrations will enhance the clarity of the methodology. Additionally, you mentioned the extraction of volatile oils without providing details about the extraction method. I recommend including supplementary information that describes the extraction method for volatile oils to ensure reproducibility.

3. The study focuses on the antiproliferative effects of Annona on five cancer cell lines: HepG2, T47D, MCF7, HCT, and Caco. It would be valuable to explain why these specific cell lines were selected and whether Annona has any effect on other types of cancer. Consider expanding the study to include additional liver cancer cell lines to strengthen the relevance of the findings. Furthermore, the evaluation of lipid peroxidation (malondialdehyde) and glutathione reductase levels is conducted only in HepG2, T47D, HCT, and Caco cell lines. To provide a more comprehensive analysis, it would be beneficial to include other relevant cell lines as well. Additionally, why were wound healing tests performed only on HepG2 and Caco cell lines? To enhance the validity of the results, I suggest incorporating a Transwell migration assay to assess cell migration. Moreover, the study's investigation of Annona's antiproliferative effects based on lipid peroxidation, glutathione reductase levels, and cell migration seems limited. To strengthen the research, consider incorporating flow cytometry to examine the impact of Annona on the cell cycle or conducting Western blot experiments to assess the effect of Annona on cell proliferation-related proteins.

In conclusion, your research study provides valuable insights into the bioassay-guided differentiation process of Annona as an anti-cancer therapy. However, to enhance the clarity, comprehensiveness, and scientific rigor of your work, I recommend addressing the points raised above. By revising the manuscript accordingly, you will ensure a more robust and impactful contribution to the field. Thank you for considering these suggestions, and I look forward to reviewing the revised version of your manuscript

Comments on the Quality of English Language

The language description in this article is clear, the overall quality is high, and only a few details need to be adjusted

Author Response

Dear Reviewers:

We highly appreciate your constructive comments. We are sure that it will add value to our manuscript. We have prepared a point-by-point response hoping that will be satisfactory. Your comments are in red, and our response is in black.

Regards…

 #Find the attached file.

Reviewer 4 Report

Comments and Suggestions for Authors

The research quality of this paper is good but the English writing of this manuscript has significant problems and must be corrected and refined thoroughly 

1) line 28 "SCoT and six ISSRs" , the full name of these brief names must be given when they began to appear in the manuscript

2) Please use the correct terminology. e.g. (line 38 "a significant concentration- cytotoxic effect" that should be described as dose response relationship; line 41 " a significant IC50" should be better described as a low IC50

3) serious grammar problems throughout the manuscript (e.g. line 81-line 84; line 101)

4) quantitative data  should be given for wounding healing assay in Figure 15 and Figure 16 

Comments on the Quality of English Language

the quality of English in this manuscript must be improved 

Author Response

Dear Reviwer #4:

We highly appreciate your constructive comments which we are sure will add value to our manuscript. We have prepared a point-by-point response hoping that they will be satisfactory. Your comments are in red, our response in black.

Regards…

Round 2

Reviewer 2 Report

Comments and Suggestions for Authors

The authors did not improve the MS in response to the reviewers' suggestions. There are still numerous linguistic issues, fonts that are not standard, and plant names that are not italicized. Furthermore, the reviewer's comments were ignored, and the responses to the reviewer's suggestions were unsatisfactory. Overall, the rewritten manuscript does not improve. Hence, I cannot recommend it for publishing in this Journal.

Comments on the Quality of English Language

There are still numerous linguistic issues in this manuscript. 

Author Response

We highly appreciate your constructive comments which we are sure will add value to our manuscript. We have prepared a point-by-point response hoping that they will be satisfactory. Your comments are in red, our response in black.

Regards…

Thank you for your esteemed commentary. We have meticulously taken into account all the comments and modifications, incorporating comprehensive revisions and engaging in linguistic editing throughout the manuscript. We have undertaken the task of editing the language in its entirety. We are optimistic that it will now be deemed acceptable by your esteemed self. However, in the event that further enhancements in terms of English language are required, we kindly request additional time in order to effectuate the necessary improvements.

Reviewer 3 Report

Comments and Suggestions for Authors

Dear Author,

Thank you for submitting your revised manuscript titled "Comprehensive tools of Alkaloid/Volatile compounds-Metabolomics & DNA profiles: Role-bioassay-guided differentiation process of Six Annona sp. grown in Egypt as anti-cancer therapy" for further review. We appreciate the effort you have put into addressing the reviewers' comments and making the necessary revisions. After careful examination, we have the following feedback and recommendations for further improvement.

1. Regarding the second issue, concerning the use of three different concentrations of methanol, we find your explanation reasonable and acceptable. We also acknowledge the supplementary information you provided regarding the volatile oil extraction method, which we have observed in your revised version. We consider this supplementation appropriate and acceptable.

2. In response to the third issue, we are pleased to see that you will consider incorporating flow cytometry or Western blot analysis in future studies to investigate the effects of Annona on the cell cycle and proliferation-related proteins, respectively. However, in the current study, you have not adequately explained the rationale for selectively using different cell lines for different experiments. It is essential to provide a clear explanation in the manuscript to improve its logical flow. Selective usage of different cell lines for different experiments may undermine the persuasiveness of the study. Additionally, using a wound healing assay alone to assess cell migration ability may lead to false-positive results due to cell proliferation. To enhance the credibility of your findings, we recommend using Transwell chambers to assess cell migration ability.

We believe that addressing these concerns will significantly strengthen your manuscript. We encourage you to expand on the mentioned points and provide a more thorough explanation in the revised version of your article. Thank you for your attention to these matters, and we look forward to reviewing the updated manuscript.

Author Response

Dear Reviwer #3:

We highly appreciate your constructive comments which we are sure will add value to our manuscript. We have prepared a point-by-point response hoping that they will be satisfactory. Your comments are in red, our response in black.

Regards…

I am writing to discuss the various points that have been pointed out in the given manuscript.

To start off, I would like to thank you for the feedback regarding the second issue concerning the use of different concentrations of methanol in my manuscript. Moreover, I would like to discuss the third issue regarding the use of transwell chambers in my study. Due to current economic issues in my country with respect to the trading side as in the import and export of goods , I am unable to find the said chambers present in the country. I have contacted the company , which imports the plates used in the transwell chambers , and I have been informed that’s imports have stopped. To combat this , I have contacted foreign companies to bring in the goods. To my dismay, I was told that the process would take a minimum of 3 months add to which 20 days of experimentation. Thus, I have decided to include this section also in my future studies including the flow cytometry and western blotting to overall increase the credibility and persuasiveness in my study.

Then again I would like to end by thanking you again for the constructive criticism provided in the given review of my manuscript.